# HTLV-1 infection of donor-derived T cells might promote acute graft-versus-host disease following liver transplantation

Chuan Shen[1,4], Yiyang Li [2,4], Boqian Wang[2,4], Zhipeng Zong[1,4], Tianfei Lu[3], Nokuzola Maboyi[2], Yuxiao Deng[1], Yongbing Qian[1], Jianjun Zhang [1] ✉, Xianting Ding [2] ✉ & Qiang Xia [1] ✉

Acute graft versus host disease (aGVHD) is a rare, but severe complication of liver transplantation (LT). It is caused by the activation of donor immune cells in the graft against the host shortly after transplantation, but the contributing pathogenic factors remain unclear. Here we show that human T cell lympho-tropic virus type I (HTLV-1) infection of donor T cells is highly associated with aGVHD following LT. The presence of HTLV-1 in peripheral blood and tissue samples from a discovery cohort of 7 aGVHD patients and 17 control patients is assessed with hybridization probes (TargetSeq), mass cytometry (CyTOF), and multiplex immunohistology (IMC). All 7 of our aGVHD patients display detectable HTLV-1 Tax signals by IMC. We identify donor-derived cells based on a Y chromosome-specific genetic marker, EIF1AY. Thus, we confirm the presence of CD4+Tax+EIF1AY+ T cells and Tax+CD68+EIF1AY+ antigen-presenting cells, indicating HTLV-1 infection of donor immune cells. In an independent cohort of 400 patients, we verify that HTLV-1 prevalence correlates with aGVHD incidence, while none of the control viruses shows significant associations. Our findings thus provide new insights into the aetio-pathology of liver-transplantation-associated aGVHD and raise the possibility of preventing aGVHD prior to transplantation.

Graft-versus-host disease (GVHD) is defined as immunological damage to host tissues and organs due to the activation of donor immune cells in graft by the host's major histocompatibility complex (MHC)[1,2]. It is a type of rejection that is stronger than typical rejections, destroying the skin and mucosal barriers of the host and leading to severe infections and ultimately lethal septic shock. The disease progression can be less than two weeks from onset to mortality[3]. Generally, GVHD can be categorized into two types, namely acute (aGVHD) and chronic (cGVHD), depending on whether the disease manifests before or after

the 100th day since transplantation[4]. In contrast to the high incidence (20–80%) of aGVHD in patients receiving hematopoietic stem cell transplantation (HSCT)[5,6], the incidence is much lower for solid organ transplantations, such as liver transplantation (LT) (0.1–2%)[7]. However, the mortality of LT patients with aGVHD is as high as 80–100%[8], whereas the overall lethality of transplant recipients is 9.4% within one year[9]. The low incidence of GVHD with high mortality is presented in all kinds of solid organ transplantations. However, there is still a lack of theory to explain this issue[10].

[1]Department of Liver Surgery, Renji Hospital, Shanghai Jiao Tong University School of Medicine, Shanghai Jiao Tong University, 160 Pujian Road, Shanghai 200127, China. [2]State Key Laboratory of Oncogenes and Related Genes, Institute for Personalized Medicine, School of Biomedical Engineering, Shanghai Jiao Tong University, 1954 Huashan Road, Shanghai 200030, China. [3]Abdominal Organ Transplantation Department, Ruijin Hospital, Shanghai Jiao Tong University School of Medicine, Shanghai Jiao Tong University, 197 Ruijin Road, Shanghai 200025, China. [4]These authors contributed equally: Chuan Shen, Yiyang Li, Boqian Wang, Zhipeng Zong. ✉e-mail: zhangjianjun@renji.com; dingxianting@sjtu.edu.cn; xiaqiang@medmail.com.cn

More than 20 years ago, donor-derived T cells were identified as the primary cause of GVHD[11]. The introduction of massive donor-derived T cells contributes to the high incidence of GVHD in HSCT patients[5,6], whereas LT patients receive a tiny proportion of donor immune cells. When a liver graft is implanted, $10^9–10^{10}$ donor-derived immune cells are transferred into the recipient's body, including T cells, B cells, natural killer (NK) cells, and monocytes[12]. In some cases, these cells can be detected in the recipient's peripheral blood and bone marrow between 3 and 100 days after surgery which matches the onset time of aGVHD[13]. Any transplant that transfers viable allogeneic lymphocytes into the recipient carries the potential risk of GVHD, such as transfusion-associated GVHD[14,15].

The critical process of the development of aGVHD is the activation of T-cell receptors (TCR) on donor-derived T cells by the antigen-presenting cells (APC) with corresponding host MHC or minor histocompatibility antigen (miH) peptides[16]. Donor CD8[+] T cells alone can induce GVHD by identifying disparities of MHC class I molecules, while CD4[+] T cells induce GVHD by recognizing MHC class II molecules[10,17,18]. T cells are the main effectors causing cell death in target tissues, mediated by various cytotoxic effects[10]. As a pathogenic and T-cell-tropic human retrovirus, human T-cell lymphotropic virus type I (HTLV-1) can infect and immortalize CD4[+] T cells[19,20]. In different animal models of HTLV-1 infection, established in macaques, rabbits, and humanized mice, the virus can cause lethal CD4[+] T-cell expansion[21–23]. Of note, the bone marrow-liver-thymus (BLT) mouse model was prone to aGVHD, related to elevated levels of HTLV-1 RNA in blood[21]. HLA matching is performed before HSCT to avoid massive donor-derived T cells attacking the hosts, while HLA matching is not mandatory in liver transplantations. Generally, the proportion of donor lymphocytes in LT is too small to induce aGVHD even though HLA is mismatching between the donors and the recipients. Additionally, HLA miss-matching did not correlate with LT outcome or

rejection rate[24,25]. In HSCT, tissue damage and GVHD seem to be the reciprocal causation[26–28]. However, in LTs, without tissue damage caused by conventional chemotherapy, the risk factors that trigger and induce the expansion of donor-derived T cells for aGVHD development remain unknown, which is our focus.

In this study, we hypothesize that viral infection plays a role in expanding donor-derived T cells in LT, which might have been masked by the severe fungi and bacteria infections in the course of aGVHD. From 2015 to 2020, seven aGVHD patients that were DCD (donor after circulatory death) recipients had been identified among 3763 liver transplant recipients in our center. Therefore, the incidence of aGVHD is 0.18% (7/3,763). To reveal the cryptogenic infections associated with developing aGVHD, we design a cocktail of probes targeting eight non-hepatotropic viruses. As a result, HTLV-1 infections are explicitly found in all the seven aGVHD patients. Furthermore, the multiplex immunohistology (IMC) results confirm that donor-derived CD4[+] T cells are increased by HTLV-1 infection. Additionally, joint analysis of mass cytometry (CyTOF) and IMC results suggest the aGVHD-specific Tax[+]CD68[+] APCs take up HTLV-1 and transmit the virus and immune signals. Thus, our study indicates that HTLV-1 is involved in the entire process of aGVHD.

## Results

### HTLV-1 only found in aGVHD patients after liver transplantation

To reveal the cryptogenic infections associated with the development of aGVHD, TargetSeq was adopted to detect viral infection in samples from two patients (ID139 and ID141) in the aGVHD group and 17 recipients in the control group (Fig. 1A). Detailed patient information and grouping principles were described in the Method and materials (Table S1). In the aGVHD group, the peripheral blood, skins, or liver biopsy tissues from only two patients were acquired due to limits of sample collection (Fig. 2A). Likewise, peripheral blood and

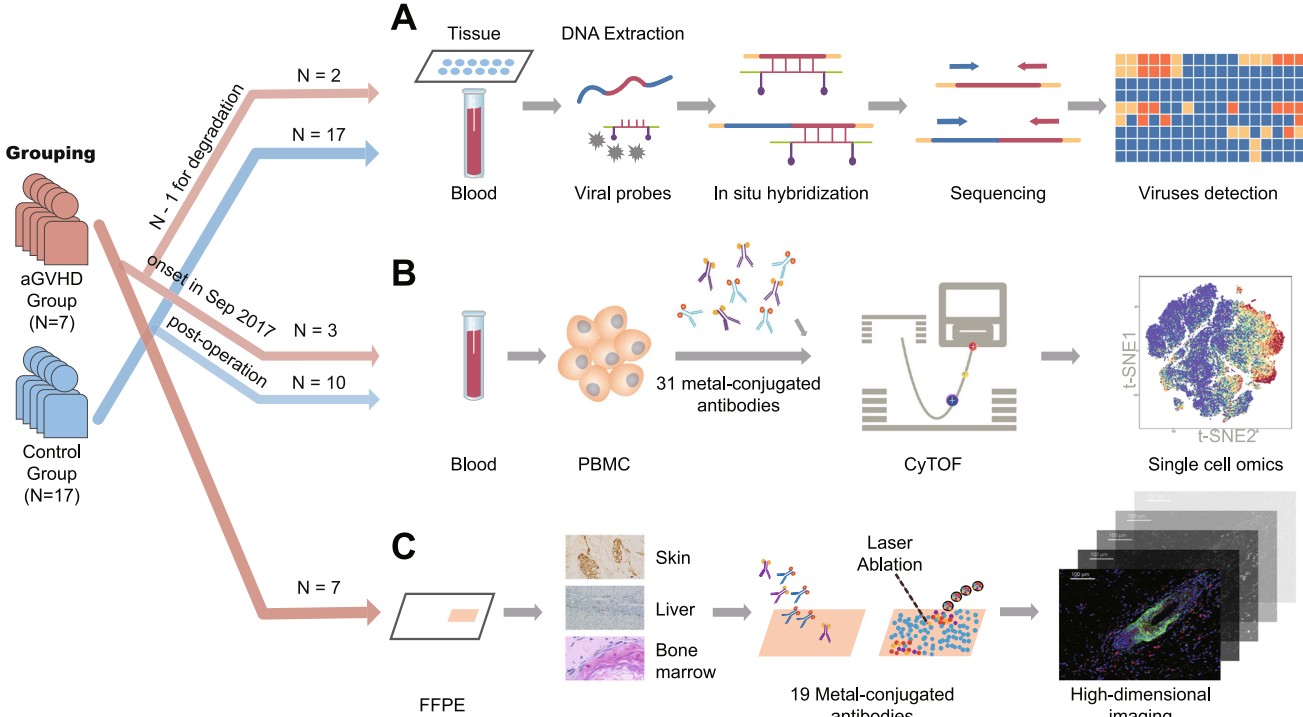

**Fig. 1 | A schematic of the experimental design. A** Multiple viral DNA and RNA sequences were detected in 37 specimens from the patients in the aGVHD group (*N* = 2) and the control group (*N* = 17). Tested specimens included skins with rash, peripheral blood, liver grafts, intestines, ventricular walls, and lungs. **B** CyTOF was performed on a total of 16 samples of peripheral blood mononuclear cells (PBMCs) from the patients in the aGVHD group (*N* = 3) and the control group (*N* = 10). **C** 17 formalin-fixed and paraffin-embedded (FFPE) samples from the patients in the aGVHD group (*N* = 7), including seven skin tissues, five native liver tissues, three graft liver tissues, and two bone marrow tissues, were assessed with IMC.

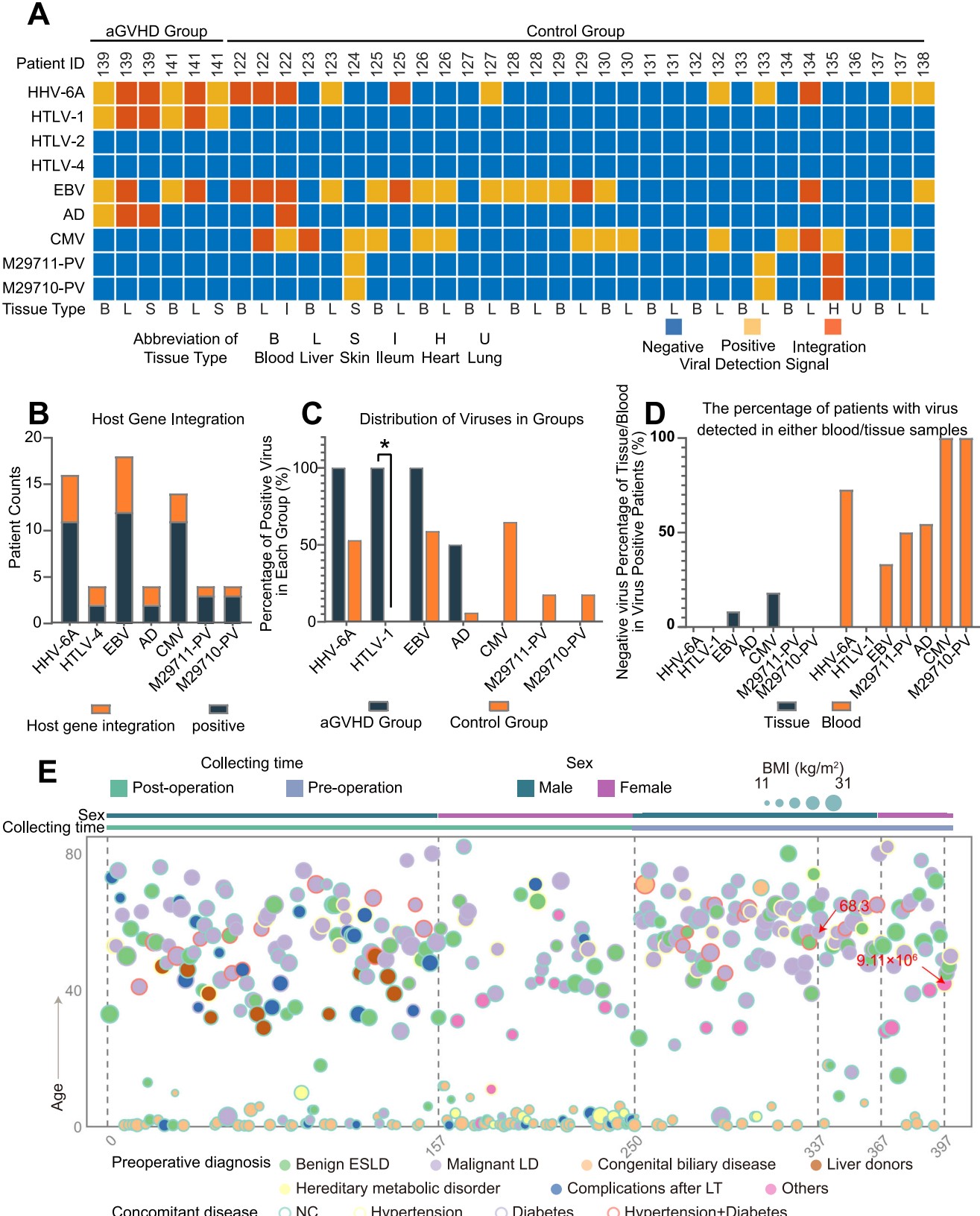

disease-related lesion tissue specimens, including liver tissues, infectious site tissues, and lung metastases of liver tumors, were collected from the control group (Fig. 2A and Table S1). TargetSeq allowed us to determine the existence of viruses (HHV-6A, HTLV-1, HTLV-2, HTLV-4, EBV, adenovirus, CMV, and parvovirus-1), especially whether viruses were in the latent infection phase[29]. Latent infection, where viral DNA

had been integrated into the host genome, is a way for viruses to achieve persistent and lifelong infection in humans and participate in the pathogenesis of various complications[30]. In both the aGVHD and control groups, HTLV-2 and HTLV-4 were not detected and therefore excluded from subsequent analysis (Fig. 2A). HHV-6A and HTLV-1 were detected in all the aGVHD samples tested, while EBV and adenovirus

**Fig. 2 | Multiple viral cDNA probe detections indicate that HTLV-1 presents exclusively in all the peripheral blood and tissues of aGVHD patients. A** The heatmap depicts the detection of virus or virus/host integration in each sample from two patients in the aGVHD group and 17 patients in the control group. Negative, positive, and host integration results are marked as blue, yellow, and red, respectively. **B** Patient numbers of virus infection and host integration were counted. **C** The percentages of virus infection rates in each group were calculated and compared between the aGVHD group ($N = 2$) and control group ($N = 17$). All $p$ values were calculated using Fisher's exact test and corrected with Benjamini-Hochberg adjustment. *$p = 0.0406$. **D** The percentage of patients with virus detected in either peripheral blood or tissue samples. **E** The heatmap depicts basic clinical information of the 400 patients in the validation cohort, which includes age, body mass index (BMI), sample collection time, pre-operative diagnosis, and concomitant diseases. The two HTLV-1 infection patients are marked by red arrows with annotations of the viral loads. (Benign ESLD: benign end-stage liver disease, Malignant LD: malignant liver disease, Complications after LT: complications after liver transplantation, NC: no concomitant disease.).

were detected in some of the aGVHD samples (Fig. 2A). Viral host gene integrations of HTLV-1, HHV-6A, and adenovirus were found in both the liver biopsy tissues and the skin tissues from patient 139, and EBV integrations in liver biopsy tissues only. Meanwhile, viral host gene integrations of HTLV-1, HHV-6A, and EBV were detected in only the liver biopsy tissues of patient 141 (Fig. 2A, B). The presence of viruses was compared between groups by Fisher's exact test and corrected with Benjamini-Hochberg adjustment (Fig. 2C). Compared with the control group, the percentage of HTLV-1 positive patients was significantly higher in the aGVHD group (adjusted $p$ value = 0.0406) (Fig. 2C). Parvovirus-1 and CMV were not detected in any of the aGVHD samples. In addition, HTLV-1 was not detected in the peripheral blood or tissue samples from the control patients, which contrasts with the aGVHD group where this virus was detected in all the samples (Fig. 2A, C, D). Although HHV-6A was detected in all samples from the aGVHD group, HHV-6A was also present in the control group. There was no significant difference in the HHV-6A infection rates between the aGVHD and control groups (Fig. 2A, C). Therefore, these results preliminarily indicated the association between HTLV-1 infection and the occurrence of aGVHD. In addition, some of the viruses were not detected in the peripheral blood of the patients with tissue infections, which suggests it may be more reliable to perform virus screening on both tissue and peripheral blood (Fig. 2D).

To further investigate the prevalence of HTLV-1 in our center and verify our hypothesis of HTLV-1 infection and aGVHD process, an independent cohort of 400 patients, including liver surgery patients, recipients, and donors of liver transplants (Table S2), were recruited from January 2021 to June 2021 for RT-PCR detection of HTLV-1 infection. A total of two patients were found HTLV-1 positive, including one donor of transplantation and one patient after partial hepatectomy (Fig. 2E). Notably, the recipient of the HTLV-1 positive donor was not infected due to the viral load of the donor being lower than the lowest load that can cause viral infection ($9 \times 10^4$)[31]. The other HTLV-1-positive patient with co-infection with hepatitis C died on the second day after surgery for acute liver failure. None of the 400 patients in this cohort was diagnosed with aGVHD, and there was no HTLV-1 positive recipient. Therefore, the prevalence of HTLV-1 in our center in patients without aGVHD was 0.5%, with a 95% confidence interval of [0.01%, 1.93%], which matched the reported prevalence of 0.1% to 1%[32]. Given that, the possibility that the co-occurrences of the two rare diseases were a coincidence was less than $10^{-4}$ using a binomial probability calculation, suggesting that HTLV-1 infection should be closely related to aGHVD development after LT.

## aGVHD-specific circulating immune cells (Tax⁺CD68⁺) uncovered in peripheral blood with features of APCs

Since the relationship between HTLV-1 and aGVHD is statistically supported, a series of explorative experiments were conducted to uncover the underlying biological mechanisms further. Mass cytometry (CyTOF) was adopted for systematic and deep phenotyping of the PBMCs from the aGVHD ($N = 3$) and control ($N = 10$) patients (Fig. 1B). Samples were collected from aGVHD patients at two different time points, and once from control patients. A CyTOF panel, including 31 metal isotope-tagged antibodies (Table S4), was designed to acquire a global overview of the leukocytes in peripheral blood. The panel contained 30 lineage markers to distinguish the significant leukocyte subsets and a marker to trace HTLV-1 infection at the cellular level. Mainly, a metal-labeled antibody against Tax protein, an HTLV-1 specific RNA transcription enhancer[33,34], was included in the panel. To characterize the phenotypes of immune cell subsets, t-distributed stochastic neighbor embedding analysis was used to generate a two-dimensional map in Cytofkit[35]. For each sample, at least $5 \times 10^3$ leukocytes (CD45⁺ cells) were selected for subsequent data analysis (Fig. S1). Results showed that the Tax protein signal was positive in all six aGVHD samples while negative in all 10 samples from the control group. The positive/negative ratio of the signal intensities was up to 50 (Fig. 3A).

In Fig. 3B, every dot represents a single cell, and color denotes the expression level of Tax protein. The spatial distributions of the immune cells were different between the aGVHD and control groups, suggesting different immune responses in these patients (Fig. 3B). We then clustered immune cells into 30 clusters based on the surface marker expressions with PhenoGraph[35] (Fig. 3C). The major different clusters between the aGVHD and control groups were Tax⁺CD68⁺ clusters (Fig. 3B and S2). In accordance with the result of TargetSeq that HTLV-1 infection should be closely related to aGHVD development after LT, Tax⁺ cells were only detected in cells from the aGVHD group, which composed the clusters 22–26# and 29-30# (Fig. 3B, C). The cell phenotypes of the Tax⁺ clusters are listed in Table S6. Cluster 23#, 25#, 29# and 30# were classified as CD3⁻CD19⁻CD11c⁺ CD123⁻ cells. Among them, Cluster 29# were HLA-DR⁺ cells, which can be defined as conventional dendritic cells (cDC cells). Dendritic cells (DC), constituting the mononuclear phagocyte system (MPS), are considered to be the most potent APCs[36]. Because CD68 is a selective marker for human monocytes and macrophages, Clusters 22#, 23#, and 30# (CD3⁻CD19⁻CD14⁻CD16⁺HLA-DR⁻CD68⁺) were considered non-classical monocytes[37]. Cluster 25# was intermediate monocytes (CD3⁻CD19⁻CD14⁺CD11c⁺CD16⁺CD68⁺). Cluster 24# (CD3⁻CD19⁻CD14⁻CD16⁻CD11c⁻CD123⁻HLA-DR⁻CD68⁺) expressed activation markers such as CD38[38] and CD69[39], indicating these cells were highly activated. Meanwhile, cluster 26# (CD19⁻CD14⁻CD3⁺CD8⁺Tax⁺) was characterized as Tax⁺CD8⁺T cells (Fig. 3C, D, and S2). PhenoGraph clustered 10 to 100 times more cells from the aGVHD group than the control group into the seven Tax⁺ clusters (Fig. 3E). The percentages of the clusters 22#, 26#, and 29–30# exceeded 5%, occupying high proportions in leukocytes in the aGVHD samples. DCs, monocytes, and macrophages have long been identified as antigen-presenting cells in circulation and in tissues[40–42]. APC, reacted with antigen, presented MHC II groove-bound peptides to CD4⁺ T cells that share the same allele, provoking immune responses[43]. Our CyTOF results revealed aGVHD-specific immune cell subsets, especially Tax⁺CD68⁺ APCs.

## "Triple-positive T cells" confirmed to be donor-derived T cells in skin sections

Following the analysis of Tax⁺ immune cell subsets in peripheral blood, IMC was further applied to study the spatial distribution of the Tax⁺ immune cells in the skin lesions, which is one of target organs in aGVHD (Fig. 1C). An IMC panel of 19 metal isotope-tagged antibodies was designed to acquire a global overview of the immune cells in the skin lesions and reveal the interactions between HTLV-1 and immune

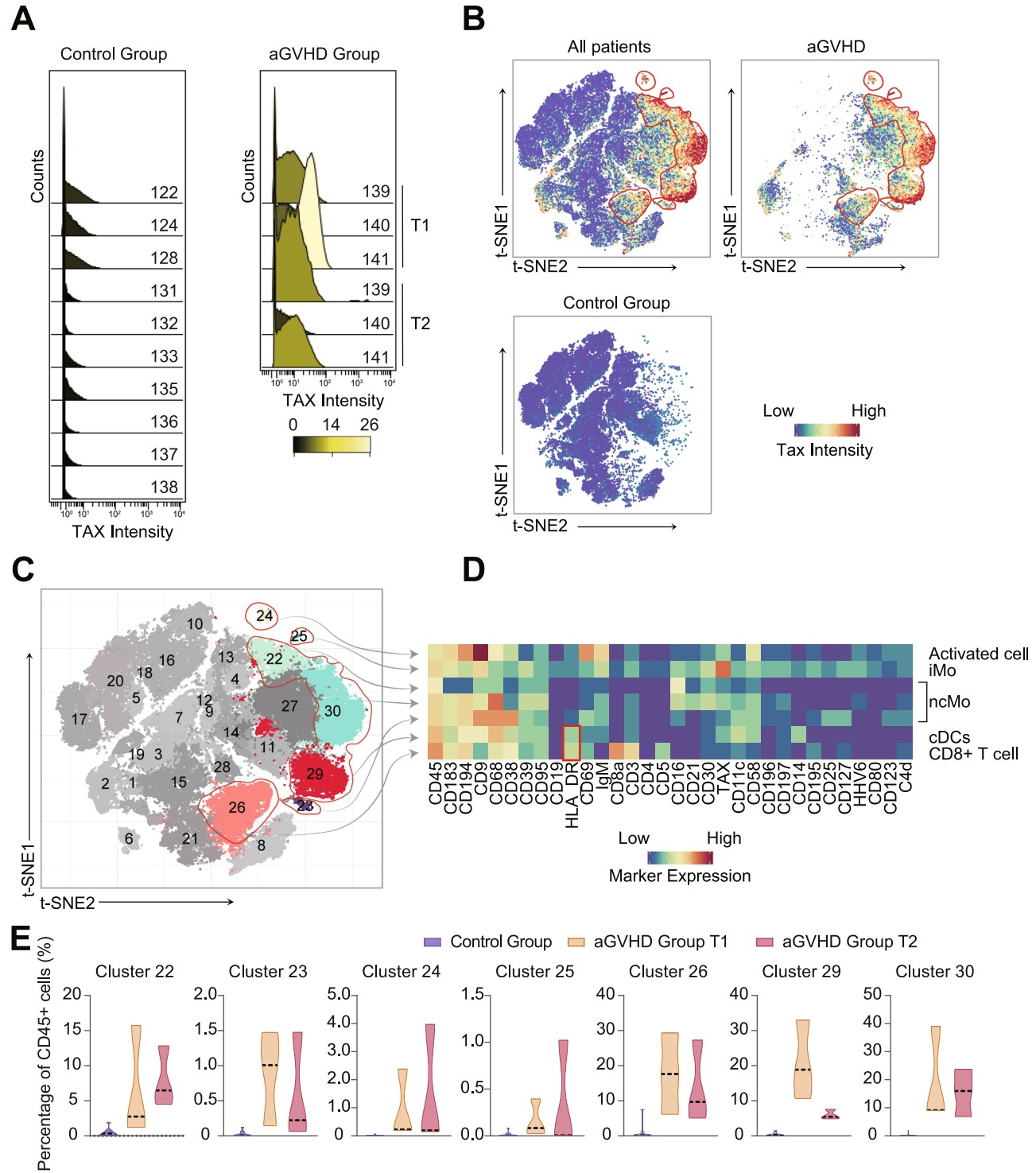

cells (Table S5). The panel contained two structure protein markers, 13 immune cell markers, an HTLV-1 specific marker (Tax protein), and a Y chromosome marker to track the source of cells. IMC was performed on skin lesions from all seven patients in the aGVHD group. Among them, 4 female patients (ID139, ID141, ID143, and ID144) received livers from male donors. Thus, eukaryotic translation initiation factor 1 A Y-linked (EIF1AY), expressed by the corresponding gene located on Y chromosome, was targeted to track donor-derived cells in samples from female recipients[44]. The signals of EIF1AY were shown on samples from female samples (Fig. 4A–D) but not on male samples (Fig. 4E–H). In addition, blood serum from 2 aGVHD patients (ID144 and ID145) and 2 control patients were also subjected to ELISA test, and the HTLV-1

antibody titers were 30 to 100-fold higher in the aGVHD samples than in the control samples (Fig. S3).

The interactions between various immune cells and HTLV-1 in skin tissue sections are shown in Fig. 4. The region of interest (ROI) was defined as the lymphocyte-rich area in the epidermal–dermal junction selected by optical microscopy. In the specimen sections from every patient in the aGVHD group, Tax protein signals were detected in the nucleus, cytoplasm, and the extracellular space (secreted protein), confirming the presence of the HTLV-1 viral protein in targeted organs, including skin and liver. In addition, Fig. 4B shows the overlay of CD4, EIF1AY, and Tax protein signals. The triple overlap signals can be found in all the skin sections from female recipients (Fig. 4B–D).

**Fig. 3 | Circulating Tax⁺ CD68⁺ immune cells were uncovered distinctively in aGVHD peripheral blood, mainly DC and monocytes. A** The two histograms visualize the intensities of HTLV-1-specific Tax signal in CD45⁺ populations (Fig. S1) from the two groups of PBMC samples. In aGVHD group, PBMCs from three aGVHD patients were collected at two-time points, T1 and T2, between onset and death. Ten patients in the control group received liver transplants and their PBMCs were collected at one postoperative time point when the complication occurred. PBMCs from aGVHD patients show high levels of HTLV-1 Tax expression. **B** t-SNE dimensional reduction was performed on CD45⁺ cells to explore the heterogeneity across patients. The color scale indicates the arcsinh-transformed signal intensity of Tax. The clusters with positive Tax expressions are marked through manual gating (red circles), primarily presented in the aGVHD group. **C** PhenoGraph and t-SNE depict the clustering result with 30 clusters identified automatically. The aGVHD group displays distinct cluster distributions from the control group. The Tax⁺ sub-populations are marked with hollow red circles (Table S6). **D** A heatmap depicts the median expression of cell markers in the seven Tax⁺ subpopulations. The HLA-DR⁺ subpopulations are marked by a hollow red rectangle. The color bars indicate arcsinh-transformed signal intensities of the proteins. **E** Percentages of the seven Tax⁺ subpopulations in CD45⁺ cells were compared between the aGVHD and control groups. Tax⁺ cells are presented mainly in the aGVHD group. (Intermediate monocyte: iMo; non-classical monocyte: ncMo; Conventional dendritic cell: cDC; CD8⁺ T cell.).

The Tax⁺CD4⁺ cells also expressed CD3, so Tax⁺CD4⁺ cells were defined as CD4⁺ T cells (Fig. S4, S5). The overlay of CD4, EIF1AY, and Tax protein signals revealed that the effector T cells of aGVHD was "triple-positive T cells", which were donor-derived and HTLV-1-infected CD4⁺ T cells. Fig. 4C, G demonstrate abundant C4d expression in immune cells that infiltrated skin lesions and show co-expression of Tax and C4d protein. Activated by the antigen-antibody reaction, C4d is an important marker in antibody-mediated humoral rejection after organ transplantation and also been reported to present after the occurrence of GVHD[45–47]. It might be a sign of interaction between viral infection and immune rejection during aGVHD progression[48]. Additionally, the overlapped signals of CD68, Tax, and EIF1AY in Fig. 4D, H implied the presence of donor-derived APCs, which is consistent with published literatures[49–51]. Furthermore, our results indicate that Tax expression was not detected in CD8⁺ T cells or B cells in the skin (Fig. S6). Single signal images of the 12 lineage markers were presented in Fig. S4, S5, S7, and S8. Thus, by labeling EIF1AY and Tax, IMC results reproduced the interaction between HTLV-1 and other immune cells, especially donor-derived cells, during the pathogenesis of aGVHD.

To fully display the HTLV-1 detection result assessed with multiple methods in the discovery cohort, the positive rate and specific detection information of HTLV-1 are listed in Table 1 and Table S1. By tracking the clinical records, we integrated the PCR detection results of HHV-6A, EBV, adenovirus, CMV, and parvovirus-1 in all the 24 patients in the discovery cohort and performed statistical analysis between the two groups (Table 1). By Fisher's exact test and corrected with Benjamini–Hochberg adjustment, significant differences in positive rate of HTLV-1 detection were found between the control and aGVHD groups (adjusted $p$ value <0.0001). The four methods cross-validated that all and only the seven aGVHD patients were infected with HTLV-1.

## CD68⁺ mononuclear phagocyte subsets defined in skin sections of aGVHD patients

After identification of Tax⁺ CD68⁺ cells in peripheral blood and skin sections, the characteristics of APCs in skin lesions of all aGVHD patients were investigated by IMC to further reveal the role of HTLV-1 infection in aGVHD. The primary APCs in the skin are tissue mono-nuclear phagocyte (MNP) subsets, which are two prominent families, DCs and tissue-resident macrophages[40]. DCs are further divided into conventional DC (cDC) and plasmacytoid DC (pDC). cDCs, pDCs, monocyte-derived macrophages (MDM), and tissue-resident macrophages respectively express CD11c⁺HLA-DR⁺CD68⁺, CD123⁺HLA-DR⁺CD68⁺, CD14⁺CD11c⁻CD68⁺, and CD14⁻CD11c⁻CD68⁺ cells[52,53]. The phenotypes and distributions of the four MNPs found by IMC in the skin lesions were presented by three representative patients (ID139, ID140, and ID145, Fig. 5A–C). Single color images of HLA-DR, CD14, CD123, CD68, and CD11c from the seven GVHD patients were shown with nuclear signals in Fig. S7 and S8. The signals of DNA, HLA-DR, CD14, CD123, CD68, CD11c, and the merged panels of these signals are shown in Fig. 5A–C. The summarized MNP phenotypes of all seven aGVHD patients are shown in Fig. 5D. Macrophages existed in skin sections of all the seven aGVHD patients, and MDMs were found in four patients (ID139, ID141, ID 142, ID144), while cDCs and pDCs only presented in skin sections of two patients (ID140 and ID143). In addition, some of the MNPs expressed the Tax protein (Fig. 4D, H), suggesting that they take up HTLV-1 and transmit the virus and the signal outward. Thus, MNP cells found in aGVHD patients were diverse, mainly DCs and macrophages, and some of them were Tax⁺CD68⁺ cells, consistent with the aGVHD-specific cell subsets in peripheral blood. The IMC results of skin sections reproduced the antigen-presenting responses and presented the APCs which were previously detected in peripheral blood and possibly migrated into the tissues.

## HTLV-1 specific signals in liver sections indicated HTLV-1 was either donor-derived or pre-existed in the recipients

To trace the source of HTLV-1, we further performed IMC to detect Tax protein expression in the liver graft and native liver of recipients. In the liver biopsies (liver graft) from female patients who received the organ from male donors (ID139), the Tax protein was detected in CD68⁺ EIF1AY⁺ cells, which confirmed donor-derived HTLV-1 infection from the liver graft (Fig. 6A, 4D and S9). In the native liver (ID139, ID144, and ID145), the Tax protein was detected on CD68⁺ cells in the native liver section, indicating the presence of pre-existing HTLV-1 infection in these patients (Fig. 6A). For both patients ID140 and ID141, the Tax protein signal was positive on the liver grafts (Fig. 6A) but negative on the native livers (Fig. 6B), suggesting HTLV-1 infection after surgery, probably donor-derived. HTLV-1 infection was not identified in the bone marrow sections or other liver tissues (Fig.6B). These results preliminarily revealed the viral infection pathways, either donor-derived or recipient pre-existed.

## Discussion

In this study, we detected HTLV-1 cDNA and HTLV-1 specific Tax proteins from peripheral blood samples, fresh tissue sections, and formalin-fixed and paraffin-embedded (FFPE) tissue sections of all aGVHD-diagnosed patients by pre-synthesized TargetSeq, CyTOF, and IMC. The association between HTLV-1 infection and the occurrence of aGVHD has been determined by statistical analysis. Furthermore, CyTOF uncovered aGVHD-specific circulating Tax⁺CD68⁺ immune cells in peripheral blood, including monocytes and DCs. Integrated analysis of CyTOF and IMC results reproduced the antigen-presenting responses during aGVHD after LT. Some MNPs, mainly DCs, MDMs,

## Table 1 | Summary of virus detection results and joint analysis

| Virus | Number of cases (infected/total) | |
|---|---|---|
| | Control group | aGVHD group |
| HHV-6A | 9/17 | 7/7 |
| HTLV-1 | 0/17 | 7/7*** |
| EBV | 10/17 | 3/7 |
| Adenovirus | 1/17 | 1/7 |
| CMV | 11/17 | 2/7 |
| parvovirus-1 | 3/17 | 0/7 |

The number of infected cases of the aGVHD group was compared with the control group by Fisher's exact test and corrected $p$ values by Benjamini-Hochberg adjustment. (***$p$ < 0.00006).

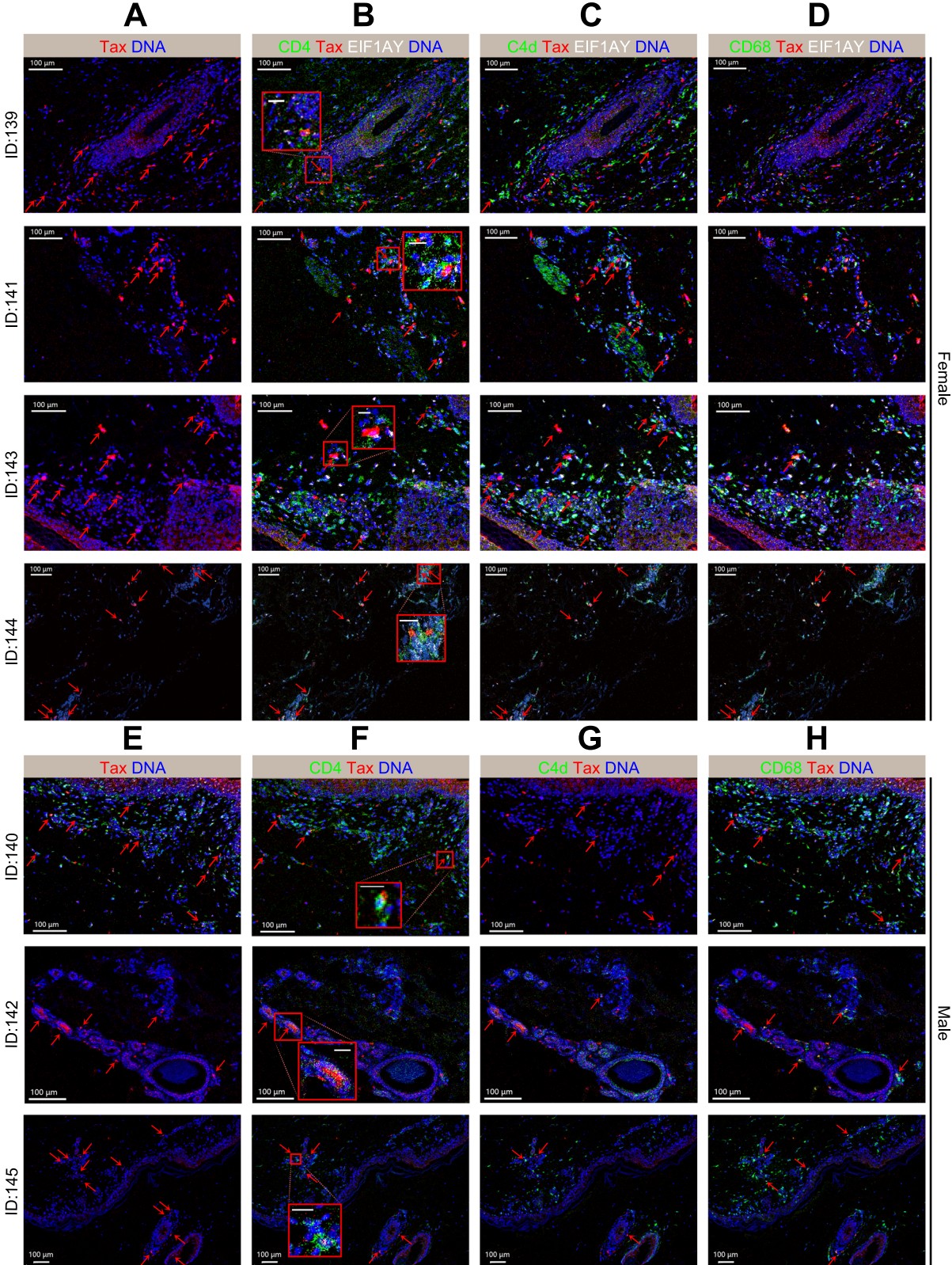

**Fig. 4 | HTLV-1 specific Tax protein detected on the aGVHD skin tissues by IMC.** Representative IMC images of two independently stained skin tissues from 7 aGVHD patients show the overlap of Tax (red), DNA (blue) and different immune response markers. Displayed channels for each column were: **A** Tax (red)/DNA (blue); **B** CD4 (green)/Tax (red)/EIF1AY (white)/DNA (blue); **C** C4d (green)/Tax (red)/EIF1AY (white)/DNA (blue); **D** CD68 (green)/Tax (red)/EIF1AY (white)/DNA (blue). **E** Tax (red)/DNA (blue); **F** CD4 (green)/Tax (red)/DNA (blue); **G** C4d (green)/Tax (red)/DNA (blue); **H** CD68 (green)/Tax (red)/DNA (blue). Red arrows pinpoint the multi-signals overlap position and red boxes show the magnified view of CD4⁺T cells with Tax signal. Scale bar = 100 µm for landscape view and scale bar = 20 µm for magnified view.

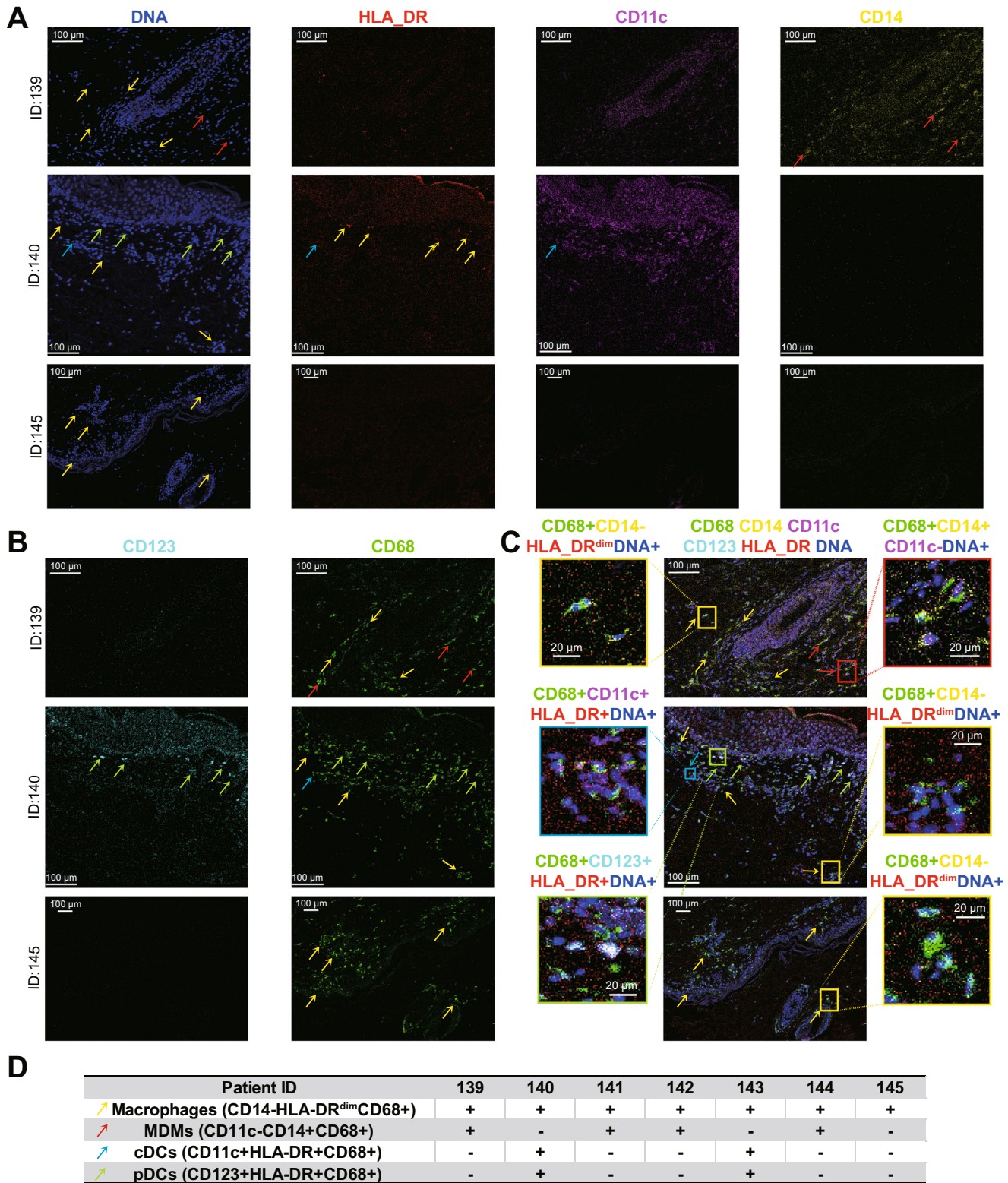

**Fig. 5 | CD68⁺ mononuclear phagocyte (MNP) in skin sections of aGVHD patients.** Representative IMC images of two independently stained skin tissues from 3 aGVHD patients (ID139, 140, and 145) show the seven single signals of **A** DNA (blue), HLA-DR (red), CD11c (purple), CD14 (yellow), **B** CD123 (light blue), CD68 (green). **C** overlay signal of seven markers. The same markers are shown in images in each column. Red, yellow, green, and blue arrows, respectively, pinpoint the monocyte-derived macrophages (MDM), macrophages, plasmacytoid dendritic cells (pDC), and conventional dendritic cells (cDC). Red, yellow, green, and blue boxes respectively show the magnified multi-signal overlap position of MDMs, macrophages, pDC, and cDC. Scale bar = 100 μm for landscape view and scale bar = 20 μm for magnified view. **D** Each column is mononuclear phagocyte (MNP) identification in skin sections from each patient. Each row is the comprehensive phenotyping of one MNP, and the symbol+/− indicates the MNP is existence/absent in skin sections of each patient. Displayed channels for each row are: macrophage (CD14⁻CD11c⁻CD68⁺); MDM (CD14⁺CD11c⁻CD68⁺); cDC (CD11c⁺HLA-DR⁺CD68⁺); pDC (CD123⁺HLA-DR⁺CD68⁺).

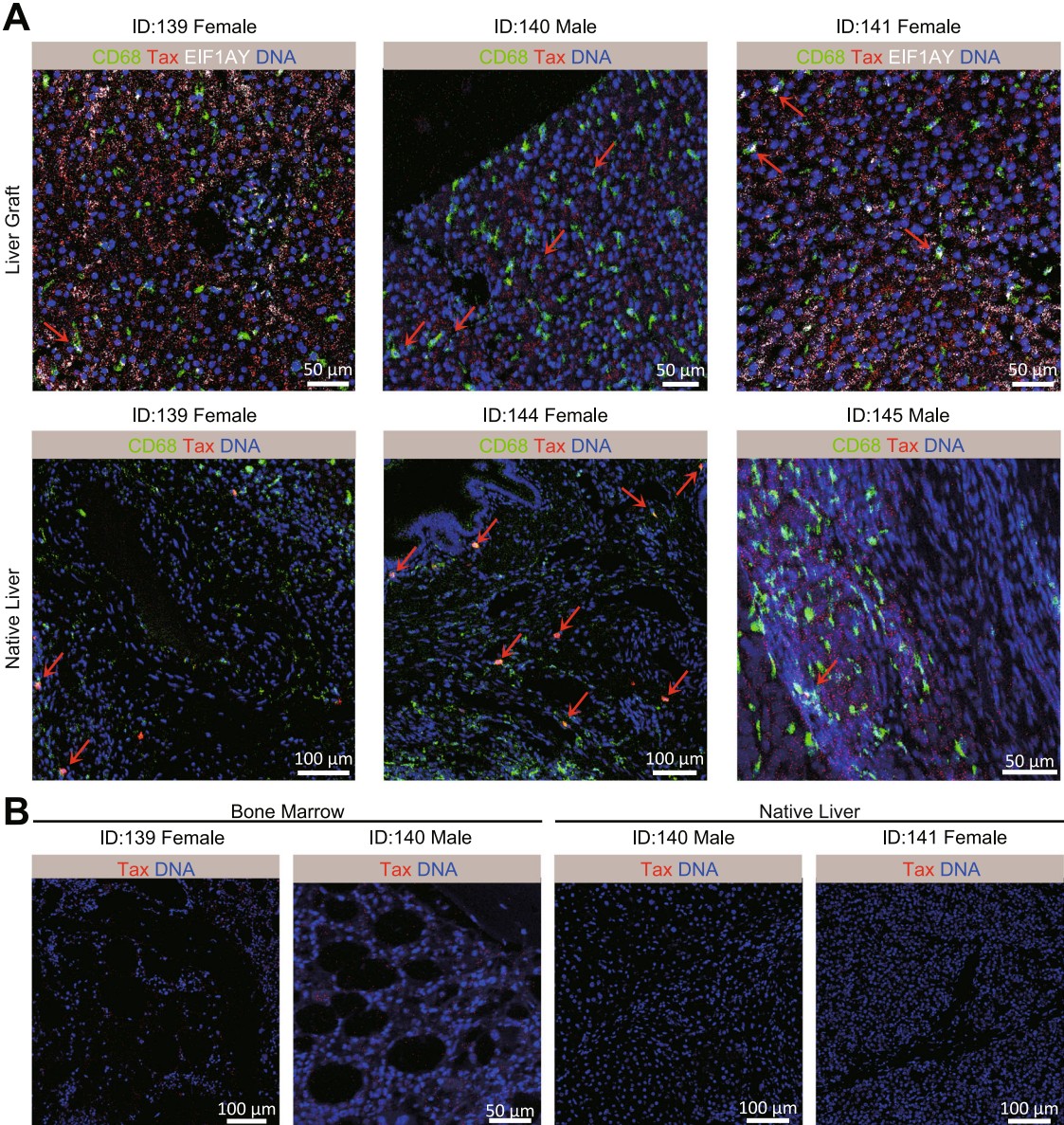

**Fig. 6 | HTLV-1 specific Tax protein detected on liver graft, native liver, and bone marrow samples from aGVHD patients by IMC. A** Representative Tax[+] images of two independently stained liver graft and native liver show two overlap patterns. Each row of figures display the same tissue type. Displayed channels for liver graft samples from left to right are: CD68 (green)/Tax (red)/EIF1AY (white)/ DNA (blue), CD68 (green)/Tax (red)/DNA (blue), CD68 (green)/Tax (red)/EIF1AY (white)/DNA (blue). Displayed channels for native liver samples are CD68 (green)/ Tax (red)/DNA (blue). Red arrows pinpoint the multi-signal overlap position. **B** Representative Tax[-] IMC images of two independently stained bone marrow and native liver sections show an overlap of Tax (red) and DNA (blue). Scale bar = 50 or 100 μm, shown in the images, respectively.

and macrophages, in the skin co-expressed Tax and CD68 proteins. It explains the migration of APCs from peripheral blood to skin lesions, during which they took up HTLV-1 and transmitted the viruses and immune signals. Most importantly, we discovered the pathogenic host-targeting T cells by IMC, which were "triple-positive T cells". Our results confirmed the existence of donor-derived CD4[+] T lymphocytes in targeted recipient organs, which were possibly expanded by HTLV-1 infection, as a potential initiator of aGVHD after LT. These findings associate the incidence of aGVHD with a viral infection, providing new perspectives for its prevention and treatment.

HTLV-1 is a highly pathogenic virus similar to HIV, but due to its geographic distribution and long latency period, HTLV-1 has not been adequately heeded[20]. In contrast to HIV causing T-cell death, HTLV-1 can efficiently infect T cells, sustain their growth, and immortalize them through Tax protein-inducing kappa B-specific proteins[19,20].

Expression of the HTLV-1 Tax protein induces enhancement of T-cell promoting factors, IL-2, IL-4, IL-6, T-cell telomerase, and Tax/rex mRNA, and reduction of T-cell apoptotic factors, such as Bim (a pro-apoptotic protein), leading to excessive proliferation of T cells (generally CD4[+] T-cell) and unlimited expression of the IL-2R alpha and IL-2 genes[19,20]. Unlimited proliferation of T cells, caused by HTLV-1 infection, acts on the human immune system and becomes an enhancer of rejection[54,55]. Therefore, HTLV-1 virus infection may expand T cells, possibly for both hosts and donors, resulting in a potential tendency for host-to-graft and graft-to-host rejections.

The confirmation and prevention of HTLV-1 infection depend on the perioperative preparation of transplantation. Viral infection can occur when the number of cells containing the provirus is more than $9 \times 10^4$ [31]. Due to the low infection rate and geographical limitations of HTLV-1, many transplantation centers do not routinely screen for

HTLV-1, resulting in overlooked viral infection. In our study, expansion of donor T cells occurred regardless of the HTLV-1 infection source (donor-derived, pre-existing, or transfusion). The lethal CD4$^+$ T cells expansion is supposed to underlay aGVHD onset and progression. Therefore, our data suggest that donors, recipients, and blood supply sources should undergo strict testing for HTLV-1 infection before surgery to prevent the disordered proliferation of T cells in organ recipients. In this study, the specificity of HTLV-1 virus infection in aGVHD patients after LT was preliminarily revealed, so a more extensive multicenter study to examine the relationship between HTLV-1 and aGVHD will be beneficial. Simplified strategies for diagnosis of HTLV-1 infection and antiretroviral therapy will play positive roles in the prevention and treatment of aGVHD after LT. At the same time, it explains one of the various viral hazards in humans and calls for more research on virus-related lymphocytic diseases.

## Methods

### Ethics, grouping principles, and human sample harvest

This study was approved by the ethical review board of Renji Hospital, Shanghai Jiao Tong University School of Medicine (clinical trial registration number: KY2019074). All patient samples were obtained with informed consent under the supervision of IRB.

The DCD recipients were divided into two groups, the aGVHD and control groups. The aGVHD group consisted of seven patients diagnosed with aGVHD in different years between 2015 and 2020. The control group consisted of 17 recipients, including post-transplant rejection patients ($N = 4$), post-transplant infection patients ($N = 3$), post-transplant regular recovery recipients ($N = 4$), and pre-operative patients ($N = 6$). For patients with aGVHD, post-transplant rejection, or post-transplant infection, samples were collected during the disease progression. Specimens were collected before surgery from pre-operative control patients. The patient IDs, clinical information, and groupings for the patients evaluated in this study are listed in Table S1. The independent cohort of 400 patients, including liver surgery patients, recipients, and donors of liver transplants, was recruited from January 2021 to June 2021 (Table S2).

For TargetSeq assay, fresh samples, including skin affected by a rash, peripheral blood, and biopsy tissues of livers, intestines, ventricular walls, and lungs, were harvested from 2 patients in the aGVHD group (ID139 and ID141) and 17 recipients in the control group (Table S1).

For CyTOF and subsequent analysis, PBMC samples were collected from 3 aGVHD patients (ID139, ID 140, and ID 141) and 10 recipients in the control group, including post-transplant rejection patients ($N = 4$), post-transplant infection patients ($N = 3$), and post-transplant regular recovery recipients ($N = 3$). The PBMCs of the aGVHD group were collected at two different time points between onset and death. The PBMCs in the control group were collected only once.

The specimens for IMC were FFPE sections of native livers, skin lesions, liver biopsies (graft tissues), and bone marrows obtained from the 7 aGVHD patients (Table S1). The FFPE sections of native livers were collected during transplant surgeries, and the other specimens were collected during the progression of aGVHD.

### Detection of non-hepatotropic viruses by TargetSeq

Fresh samples ($N = 37$) were harvested from the aGVHD and control group patients, including skin lesions, peripheral blood, and biopsy tissues of livers, intestines, ventricular walls, and lungs (Table S1). The RNA was extracted by Pre-NAT Full-Automatic System (PerkinElmer, Massachusetts, USA). Capture probes targeting non-hepatotropic viruses (TargetSeq, iGeneTech, Beijing, China) were RNA probes based on liquid-phase chips (Fig. 1A). The targeted viruses included human herpesvirus 6 A (HHV-6A), Epstein-Barr virus (EBV), adenovirus, parvovirus-1, cytomegalovirus (CMV), HTLV-1, HTLV-2, and HTLV-4. Probe panels were designed for target region sequencing based on each virus' whole-genome sequence acquired from the NCBI GenBank

database (Table S3). For RNA viruses, the probes were designed using cDNA synthesized by reverse transcription as templates. TargetSeq was capable of capturing the full-length sequence of the target viruses and the corresponding genome integration regions.

### Detection of HTLV-1 by RT-PCR

The peripheral blood was collected and the RNA was extracted by Pre-NAT Full-Automatic System (PerkinElmer, Massachusetts, USA). Real-time RT-PCR assays for HTLV-1 RNA detection were performed using Human T-cell Lymphoma Virus 1 (HTLV-1) Probe qRT-PCR Kit (YaJi Biological, Shanghai, China) in an ABI 7500 real-time PCR system (Thermo Fisher Scientific, California, USA) following the kit instruction. Each reaction mixture contained 10 μL buffer, 2 μL enzyme mixture, 2 μL mixture of forward and reverse primer, 1 uL probe solution, and 5 μl specimen. The thermal cycling parameters were 30 min at 50 °C for reverse transcription, 10 min at 94 °C for predegeneration, and 40 cycles containing 15 s at 94 °C and 1 min at 60 °C.

### Antibody preparation

The antibodies used in this study and the corresponding manufacturers and concentrations are listed in Table S4 and Table S5. Metal-conjugated antibodies were purchased or prepared using a Maxpar ×10 antibody labeling kit (Fluidigm Sciences, San Francisco, USA). After conjugation, antibodies were stored in Candor PBS Antibody Stabilization Solution (Candor Bioscience, Wangen, Germany) at 4 °C.

### Preparation and staining of PBMCs for mass cytometry detection

PBMCs were separated by density gradient centrifugation with Ficoll-Paque PLUS (Amersham Biosciences, Piscataway, USA) (Fig. 1B). After lymphocyte extraction, cisplatin (5 μM) was used to treat cells in suspension, and the cells were then fixed in paraformaldehyde (PFA; final concentration 1.6%). Cells were stained with a cocktail of 30 metal isotope-conjugated antibodies against surface proteins (Table S4). After surface-protein staining, the cells were treated with the Transcription Factor Permeabilization kit (eBioscience, Santiago, USA) and then stained with anti-Tax antibody at room temperature for 30 min (Table S4). Nuclear were stained with 1 mL of 1:4000 diluted $^{191}$Ir/$^{193}$Ir DNA intercalator (Fluidigm Sciences, San Francisco, USA) with Maxpar Perm-S Buffer (Fluidigm Sciences, San Francisco, USA) overnight at 4 °C.

### Preparation of aGVHD tissue sections for IMC detection

Tissue sections were cut from FFPE tissue blocks of host livers, skins, bone marrow biopsies, and graft liver biopsies from Renji Hospital (Fig. 1C). The tissue slides were deparaffinized in xylene (Adamas-beta, Shanghai, China) and rehydrated in a graded ethanol (Adamas-beta, Shanghai, China) series. Then, they were incubated in a preheated retrieval solution (R&D Systems, Boston, USA) at 95 °C for 30 min and cooled to room temperature. The tissue slides were then blocked with 3% BSA (Macklin, Shanghai, China) in DPBS (Gibco, San Francisco, USA) for 45 min at room temperature. After antigen retrieval, the tissue slides were incubated with an antibody cocktail, containing 19 metal isotope-tagged antibodies (Table S5) targeting immune cell markers and Tax, overnight at 4 °C in a humidified chamber. For nuclear staining, each section was stained for 30 min at room temperature with 100 μL of 1:600 diluted $^{191}$Ir/$^{193}$Ir DNA intercalator (Fluidigm Sciences, San Francisco, USA) with DPBS.

### Mass cytometry detection and subsequent analysis

After acquisition on Helios (Fluidigm, San Francisco, USA), data were normalized with four standard EQ beads (Fig. 1B). Mass cytometry data plots and histogram analysis were performed on CytoBank's online platform (www.cytobank.org). A series of gates were used to select single cells and CD45$^+$ cells, as depicted in Fig. S1. Parameters for dimensionality reduction were set, and PhenoGraph was applied to

dimensionality reduction using the R package Cytofkit[35]. ArcSinh transformation, with a scaling factor of 5, was chosen as the transformation method to diminish the noise in these measurements.

### IMC detection and subsequent analysis

Images were acquired with a Hyperion Laser Scanning Module (Fluidigm, San Francisco, USA) coupled to Helios mass cytometer (Fig. 1C). Fluidigm's CyTOF v6.7 software-generated a.mcd file and a.txt file. The MCD Viewer v1.0 software was used for image processing and visualization.

### ELISA essay

Serum samples were collected from the patients (ID:144 and ID:145) after diagnosing aGVHD. Blood samples were centrifuged for 5 minutes to separate serum. The serum was stored at -80°C. Samples were all tested by HTLV-I ELISA Kit (KNUDI, Quanzhou, China). All operations were carried out according to the kit instructions and the optical density was measured at 450 nm using a microtiter plate reader (Bio-Tek, Winooski, USA).

### Statistical analysis

Fisher's exact test was adopted to test the differences in HTLV-1 positive rate between groups, and Benjamini-Hochberg adjustment was applied to control the false discovery rate. The adjusted $p$ value <0.05 was considered statistically significant. In Fig. S3, the differences in HTLV-1 positive rate between groups were calculated by the Mann-Whitney test.

### Reporting summary

Further information on research design is available in the Nature Portfolio Reporting Summary linked to this article.

## Data availability

The datasets of the TargetSeq and CyTOF detections of the study are available on Zenodo (https://doi.org/10.5281/zenodo.7333323 and https://doi.org/10.5281/zenodo.7333412). The raw image files are available on Zenodo (https://doi.org/10.5281/zenodo.7333438). Source data are provided with this paper.

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

## Acknowledgements

We thank professor Chih-Ming Ho (University of California, Los Angeles) for his guidance, Dr. Q. Wei and Dr. J. Zheng (Department of pathology, Shanghai 10th People's Hospital) for helping us with the immunohistochemistry, and professor Y. Ji (Zhongshan Hospital Affiliated to Fudan University) for performing the pathological diagnosis. We thank Dr. Wenqiong Su and Ms. Aiting Wang (Shanghai Jiao Tong University) for their technical assistance. This study was supported by the National natural science foundation of China (81871448). Fund beneficiary: X.D.

## Author contributions

Conceptualization, C.S. and Y.L.; Methodology, C.S. Y.L. B.W. and Q.X.; Investigation, Y.L. and C.S.; Formal Analysis, Y.L.; Writing – Original Draft, C.S. and Y.L.; Writing – Review & Editing, B.W., X.D., J.Z., Y.Q., Y.D., Z.Z. and N.M.; Visualization, Y.L.; Resources, C.S., T.L., and Z.Z.; Funding Acquisition, X.D., J.Z., and Q.X. Supervision, X.D. and Q.X.

## Competing interests

The authors declare no competing interests.
