## [Peer Review File · Nature Communications]

HTLV-1 infection of donor-derived T cells might promote acute graft-versus-host disease following liver transplantationREVIEWER COMMENTS

Reviewer #1 (Remarks to the Author):

Shen et al attempted to identify the donor type T cells that induced acute GVHD in patients of liver transplantation by PCR analysis of virus DNA in the blood and tissue samples, CyTOF analysis of T and APC subsets in the PBMC, and imaging mass cytometry (IMC) analysis of infiltrating T and APC subsets in the GVHD target tissues. The authors conclude that the HTLV-1 infection of donor-derived CD4+ T cells is a potential initiator of acute GVHD after liver transplantation. Unfortunately, the manuscript is written in a way that is very difficult to understand, the figures are presented in a very confusing way, and the indications do not appear to be supported by the data. Below are the major concerns.

Major concerns:

1. Table S1 patient info is very confusion and need to be improved. Fig. 2 labeling is very confusion and need to be simplified. The correlation analysis about HTLV-1 and acute GVHD was not clear or convincing. The description in the text needs to be improved.
2. Fig. 3 about circulating TAX+CD68+ cells with features of APC is not clear or convincing.
3. Fig. 4 IMC analysis of skin "triple-positive" CD4 T cells with expression of CD4, EIF1AY, and Tax protein is not clear or convincing. Single color and overlap need to be shown. In addition, human macrophages express CD4. To identify CD4+ T cells, CD3+CD4+ needs to be shown.
4. Similarly, indications drew from data in other figures are not convincing.

Reviewer #2 (Remarks to the Author):

Statistical Review:

Overall, I think this is an interesting and well-written manuscript describing the association between HTLV-1 infection and acute graft vs. host disease following liver transplantation. There are a few minor grammatical errors that I assume were picked up by other reviewers. I am focusing on a statistical review, only. I will note that "FFPE" is not defined.

In the Statistical Analysis section, the authors only report performing Fisher's exact tests to compare Group A to Group B and to Group C. In the Results section, regarding detection of HTLV-1 infection: Was RT-PCR used in Groups A-D to determine infection with HTLV-1? My assumption based on the reading of the methods and results is that HTLV-1 infection was determined by the methods described in the results and not using RT-PCR. Group A (the group with aGVHD) had a sample size of 3 but one of the samples (ID #140) suffered from degradation during transport and could not be assessed. HTLV-1 was detected in samples from the other two (ID #s 139 and 141). Therefore, when comparing Group A to Group B (with no HTLV-1 infections), a Fisher's exact test of 2/2 (100%) vs. 0/11 (0%) was performed, resulting in a p-value of 0.0128. I replicated this. Comparing Group A to Group C (with no HTLV-1 infections), a Fisher's exact test of 2/2 (100%) vs. 0/6 (0%) was performed resulting in a p-value of 0.0357, also replicated by me. While these probabilities are below the nominal level of 0.05, a sample size of 2 in Group A is not compelling. Additionally, a sample size of 2 is extremely unlikely to be representative of the target population. Further, there was no adjustment made for multiple comparisons. A simple and acceptable multiple comparison adjustment for this case could be a Bonferroni correction for the number of comparisons being made, in this case 2. Therefore, a Bonferroni-adjusted type I error rate for p-value comparison would be $0.05/2=0.025$. If this were used, then Group A vs. Group B would still result in a "significant" difference while Group A vs. Group C would not. What would then be the conclusion?

At the end of the first paragraph of the Results section, the authors state, "Given the overall positive rate of HTLV-1 in all tested samples was 6/37, the possibility (p-value) that it was a coincidence was less than 10^{-7} , which was practically preclusive." Please delete the last phrase ("which was practically preclusive") and note this language was used again in the first paragraph of the discussion. The word "preclusive" is not being used in an appropriate way and is not typical

language used for interpreting p-values. I spent a long time trying to understand where these numbers came from (deduced from Figure 2A). Please refer the reader to Figure 2 in this text or better describe where the numbers 6 and 37 come from. The calculation of 10^{-7} is assumed to be a binomial probability where the sample size is 37, the number positive is 6, and the probability of 1 sample being positive is 0.005 (from the prevalence of HTLV-1 in the cohort of 400 patients). I don't know what criteria is used to formally diagnose a patient with HTLV-1—is it a positive from any of possibly multiple tissue/blood samples in Groups A-D (assuming not by PCR)? At any rate, the estimated prevalence rate for HTLV-1 of 0.005 is the probability of a person being diagnosed with HTLV-1, not the probability of a single tissue or blood sample being positive for HTLV-1, which I assume would be higher since multiple tissues from the same person could yield positive results. It is not appropriate to use this value in your binomial probability calculation unless only a single tissue positive for HTLV-1 is equivalent to or highly correlated with a positive RT-PCR test (which was used for the cohort of 400 to estimate the prevalence of 0.005). Additionally, your statistical analysis section should describe using a binomial probability calculation for this assessment—I had to guess what you did and ultimately calculated a probability close to what you reported.

Describe in the Statistical Analysis section how you calculated the 95% CI for the prevalence of HTLV-1+ reported in the second paragraph of the Results. I cannot replicate this using standard methods.

In the Results subsection "Triple-positive T cells...", second half of the first paragraph, Group D results are presented for the first time with results only by ELISA (but no statistics to support conclusion). This section then refers the reader to Table 1 and reports p-values from the comparison of Groups A and B, this time including all 3 samples from Group A, but it was already established that Group A ID #140 was not viable. Unless I misunderstand the type of testing performed on the peripheral blood and liver graft samples in Group A, you already presented these results with different sample sizes (2 vs. 3). I believe this sentence should be deleted and, instead, refer the reader back to the results presented in "HTLV-1 only found in aGVHD patients..." if necessary.

The final sentence of this first paragraph concludes that the four methods cross-validated that the 7 aGVHD patients (only) were infected with HTLV-1. This is a bit of a strong statement given that testing via TargetSeq in Group A consisted of only n=2 and only ELISA was performed in n=4 in Group D. Your results would be stronger if there was an overlapping testing mechanism in Groups A and D.

In the Discussion, first paragraph, you mention "According to probability analysis, it is statistically preclusive that HTLV-1 infection and aGVHD accidentally occurred at the same time for all seven aGVHD patients." However, I don't see analysis results presented where all 7 were combined with a probability calculated like you did for Group A alone.

There are no strengths or weaknesses presented. The very small sample sizes are a limiting factor of these results. More emphasis should be placed on the need for larger studies adequately powered to detect the relationship between HTLV-1 and aGVHD.

Table 1 is problematic in that it implies that all three samples in Group A were used for the calculation of reported statistics when really only 2 were used (for TargetSeq). The p-values reported in the footnote are for 2/2 in Group A not 3/3. Consider changing Method to "TargetSeq or IMC" but then add a footnote that the p-values were calculated only from TargetSeq analysis. I don't see enough description in the text of the IMC results to understand how HTLV-1+ was determined via that method.

I note that Figure 2A reports that, in particular, Groups B2 and B3 use tissue samples not obtained in the other groups. Is it typical to compare different types of tissue samples?

Figures 2B and 2D: The y-axis in both figures should be either "Sample Counts" or "Sample Proportions", but not both. Update appropriately.

Table S1: The second column is unclear. The header ("HTLV(+/-) 1.Targetseq TM / 2. Cytof / 3.

IMC") does not sufficiently explain the content of the cells in that column (e.g., "C (skin, native liver) / HTLV(+) / 3 B/HTLV-1(+) (ELISA)").

Table S2: Just confirming that the age range of the cohort of 400 patients ranged from a minimum of 3 months (0.25 years) to 82 years. Clinically, is it appropriate to include children and infants? For the Gender, Concomitant diseases, Preoperative diagnosis, and Collecting time, I assume these are counts? Please indicate units of measure ("n").

Reviewer #3 (Remarks to the Author):

Li et al., used TargetSeq to determine the presence of viral nucleic acid in various tissues derived from transplant donors associated with acute Graft vs Host Disease (aGVHD) and conclude that there is an association between HTLV and aGVHD. They then use CyToF to show that aGVHD patients have a unique immune cell subset profile in blood and that CD68+ APCs are preferentially positive for HTLV (Tax). Importantly they go on to use imaging mass cytometry to show that skin lesions also contain CD68+ APCs (likely to have migrated in from blood) and that these cells colocalise with CD4 T cells which are also Tax+. Finally, they show CD68+ Tax+ cells in liver.

Overall, this study represents a significant paradigm shift in our understanding of aGVHD and this study is of high interest. However, many of the results are quite suggestive and not definitive in its current form. I also found this manuscript very confusing to read indeed and I was not been able to interpret the data as the authors have. There are numerous grammatical errors throughout the manuscript that should be corrected. This made it difficult to understand the manuscript in places. There is also some major misalignment of Figures and tables. For example, the authors direct the reader to Table S3 for the CyToF panle but this is in fact listed in Table S4.

Figure 2: I am especially confused by the message of Figure 2. At the beginning of the results section the authors do not tell us what Group A, B or C are and this is also not defined in the Figure legends of Figures 1 or 2. HTLV was only detected in the 6 group A donors but not in any of the 31 donors in group B or C. Basically the authors are saying that a wide assortment of viruses were detected in a wide range of tissues (inconsistent between donors) from a wide range of donors. They then say "particularly, HTLV-1 infections were detected in none of samples from the control groups but in all the samples from aGVHD patients". Where is this data – am I missing something here? They go on to say "Given the overall positive rate of HTLV-1 in all tested samples was 6/37, the possibility (p-value) that it was a coincidence was less than 10⁻⁷, which was practically preclusive". I do not understand how detecting HTLV in only 6 of 37 donors is significant? There seems to be nothing special about HTLV presented in Figure 2. I am sure I must be missing something. This section needs to be rewritten if this reviewer is going to understand it.

Figure 3: The authors show that PBMC derived from HTLV+ patients show a different immune cell profile to HTLV- patients and the many HTLV+ (Tax+) cells express CD68 which is a marker on monocytes in blood and macrophages in tissue. Although I agree with the concluding remark of the results describing the figure "Thus, CyTOF results revealed aGVHD specific immune cell subsets, especially Tax+ CD68+ APC cells" I do not agree with way the authors define cell population in blood – especially macrophages which are tissue resident cells! Furthermore, macrophages are very weak antigen presenting cells. The authors claim to have identified macrophages as CD68+CD11c+ HLA-DR+ cells in peripheral blood. This does not bode well for the authors understanding of the mononuclear phagocyte system as macrophages are not present in blood and do not express CD11c. These cells are more likely a NK or DC population.

Figure 4-6. The use of imaging mass cytometry is a clear strength of this manuscript. However, it is a shame that some key APC markers are missing. It would have been useful to include some more macrophage specific markers such as FXIIIa and also DC markers such as CD1c. There are no markers to include cDC1 (e.g. XCR1, CADM1, CLEC9A).

- It is of note that DCs and macrophages both express CD4, so this combination of marker expression cannot be used to definitively define T cells. The authors need to show these cells are negative for APC markers (e.g. CD11c, CD14, CD68 and HLA-DR also included in the panel).

- Recently, CD14+ tissue cells have been shown to consist of monocyte derived macrophages (CD14+ CD1c- CD11c-) and monocyte derived dendritic cells (CD14+ CD1c+ CD11c+). Both of these express CD68 – PMID 33846309. As DCs are more potent APCs than macrophages it would be helpful for the authors to define if the APCs they define in skin express CD11c (which is included in the IMC panel) and discern if they are MDM or MDDC.

Authors' Response to the Reviewer Comments

Manuscript ID: NCOMMS-21-40067A

Title of Paper: HTLV-1 infection of donor-derived T cells underlies aGVHD after liver transplantation

We truly appreciate the editorial board and reviewers' time and effort in reviewing this manuscript. We understand that the major concern from editors and reviewers was the initial discovery cohort sample size. Therefore, we would like to provide the following further elaborations to explain what we have done during the manuscript revision, and indicate our analysis why we would consider the correlation between HTLV-1 and aGVHD is indeed convincing.

As the largest liver transplant center in Asia, our hospital has identified a total of 7 aGVHD patients after 4,023 liver transplant surgeries during 2015-2022. The 7 aGVHD patients have been fully included in our study. While revising the manuscript, we have never stopped looking for new aGVHD patients to further increase the discovery cohort size. However, since aGVHD has low incidence (~0.1% in our center), the onset of new aGVHD patients is highly unpredictable and we have not seen new cases of aGVHD during the past 7 months to provide fresh tissue samples for TargetSeq assay. Therefore, to best address the editorial and reviewers' inquiry, we have adopted multiple cross-validation approaches to best explore the existing 7 aGVHD patients to justify our conclusions as below.

We managed to collect fresh tissue lesion samples from the first 2 aGVHD patients, while only formalin fixed and paraffin embedded (FFPE) samples were available from the other 5 patients, which were not suitable for TargetSeq assay. These patients quickly passed away after being diagnosed, leaving us no chance to further collect fresh tissues.

Thus, imaging mass cytometry (IMC) were performed on FFPE samples from all the 7 patients. IMC is also a solid detection method that can detect signal of HTLV-1 by capturing Tax protein, an HTLV-1 specific RNA transcription enhancer. With IMC examination, the sample size of the aGVHD patients in statistical tests was expanded to 7 instead of 2, which eliminated the possibility that the conclusion is accidental. In our revised manuscript, we have therefore simplified the grouping to better show how the detection of the 7 aGVHD patients were unified. The 24 recipients in the discovery cohort are now divided into two groups, namely the aGVHD group (n=7) and the control group (n=17). TargetSeq assay preliminarily indicated the correlation between HTLV-1 infection and the occurrence of aGVHD (adjusted p-value = 0.0406), and IMC is now used as major technique to conclude the correlation with a Fisher's exact test of 7/7 vs. 0/17 resulting in a p-value of 0.00001. Moreover, the HTLV-1 infected male donor-derived T cells were found in skin sections of female aGVHD patients, further enhancing the connection between HTLV-1 and aGVHD.

With these elaborations, we hope the editor and reviewers may agree that we indeed try our best to collect all available aGVHD samples and meanwhile the association between HTLV-1 infection and aGVHD development was statistically significantly re-confirmed in our study. We have also carefully studied the reviewers' other inquiries and prepared a 68-pages detailed point-to-point response letter to show our appreciation and respect for the time and efforts that the reviewers spent reading and helping on our manuscript.

We have substantially revised our manuscript based on provided comments and suggestions. The reviewers' comments are colored in **blue**; our responses are colored in **black**, and changes made to the manuscript are colored in **red**. Together with the revised manuscript, we hope this letter adequately addresses the reviewers' inquiries. We thank the reviewers for aiding us in improving the quality of our manuscript.

Response to Comments from Reviewer 1

Reviewer 1

We genuinely appreciate the reviewers' gracious comments. These expert comments indeed largely facilitate our revision and increase the readability of our manuscript. We would like to provide the following point-to-point response.

Shen et al attempted to identify the donor type T cells that induced acute GVHD in patients of liver transplantation by PCR analysis of virus DNA in the blood and tissue samples, CyTOF analysis of T and APC subsets in the PBMC, and imaging mass cytometry (IMC) analysis of infiltrating T and APC subsets in the GVHD target tissues. The authors conclude that the HTLV-1 infection of donor-derived CD4+ T cells is a potential initiator of acute GVHD after liver transplantation. Unfortunately, the manuscript is written in a way that is very difficult to understand, the figures are presented in a very confusing way, and the indications do not appear to be supported by the data. Below are the major concerns.

1. Table S1 patient info is very confusion and need to be improved. Fig. 2 labeling is very confusion and need to be simplified. The correlation analysis about HTLV-1 and acute GVHD was not clear or convincing. The description in the text needs to be improved.

Thanks a lot for the comments. We apologize for the confusion caused by Figure 2 and Table S1 in the original manuscript. A total of 7 aGVHD patients were recruited for our study and their blood and tissue samples were respectively assessed with hybridization probes (TargetSeq), mass cytometry by time of flight (CyTOF), and imaging mass cytometry (IMC). We have simplified the grouping of aGVHD patients and control patients for better understanding. A series of new statistics have been performed on the regrouped patients to make Figure 2 clear. Figure 1 and Table S1 have been adjusted accordingly. Descriptions about the analysis results of TargetSeq have been re-organized and re-written (**line 116-123, line 128-131, line 230-246, line 250-254, line 258-264, line 270-272, and line 277-280**).

Figure 1. A schematic of the experimental design. (A) Multiple viral DNA and RNA sequences were detected in 37 specimens from the patients in the aGVHD group (N=2) and the control group (N=17). Tested specimens included skins with rash, peripheral blood, liver grafts, intestines, ventricular walls, and lungs. (B) CyTOF was performed on a total of 16 samples of peripheral blood mononuclear cells (PBMCs) from the patients in the aGVHD group (N=3) and the control group (N=10). (C) 17 FFPEs from the patients in the aGVHD group (N=7), including 7 skin tissues, 5 native liver tissues, 3 graft liver tissues, and 2 bone marrow tissues, were assessed with IMC.

Figure 2. Multiple viral cDNA probe detections indicate that HTLV-1 presents exclusively in all the peripheral blood and tissues of aGVHD patients. (A) The heatmap depicts the detection of virus or virus/host integration in each sample from 2 patients in the aGVHD group and 17 patients in the control group. Negative, positive, and host integration results are marked as blue, yellow, and red, respectively. (B) Patient numbers of virus infection and host integration were counted. (C) The percentages of virus infection rates in each group were calculated and compared between the aGVHD and control groups. HTLV-1 was detected in all the 6 samples from the 2 patients in the aGVHD group, but none of the samples from the control group. All p-values were calculated using Fisher's exact test and corrected with Benjamini-Hochberg adjustment. * $p < 0.05$. (D) The percentage of patients with virus detected in either peripheral blood or tissue samples. (E) The heatmap depicts basic clinical information of the 400 patients in the validation cohort, which included age, body mass index (BMI), sample collection time, preoperative diagnosis, and concomitant diseases. Red arrows with annotations of the viral loads marked two HTLV-1 infection patients. (Benign ESLD: benign end-stage liver disease, Malignant LD: malignant liver disease, Complications after LT: complications after liver transplantation, NC: no concomitant disease.)

Table S1 Patients' information profile.

ID	Group	Sex	Age	Diagnosis	Sample type	HTLV-1 detection method				HTLV-1 detection summary
						IMC	Targetseq	CyTOF	ELISA	
145	GVHD	Male	41	aGVHD After OLT	FFPE of skin, native liver; Blood	+	NA	NA	+	+
144	GVHD	Female	62	aGVHD After OLT	FFPE of skin, native liver; Blood	+	NA	NA	+	+
143	GVHD	Female	47	aGVHD After OLT	FFPE of skin	+	NA	NA	NA	+
142	GVHD	Male	50	aGVHD After OLT	FFPE of skin	+	NA	NA	NA	+
141	GVHD	Female	54	aGVHD After OLT	Tissue/FFPE of skin, native/donor liver; Blood	+	+	+	NA	+
140	GVHD	Male	45	aGVHD After OLT	Tissue/FFPE of skin, native/donor liver; Blood	+	NA	+	NA	+
139	GVHD	Female	63	aGVHD After OLT	Tissue/FFPE of skin, native/donor liver; Blood	+	+	+	NA	+

138	Control	Male	48	Normal recovery After OLT	Tissue of donor liver; Blood	NA	-	-	NA	-
137	Control	Male	59	ACR After OLT	Tissue of donor liver; Blood	NA	-	-	NA	-
136	Control	Male	23	HCC recurrence After OLT	Tissue of lung; Blood	NA	-	-	NA	-
135	Control	Female	31	IFI After OLT	Tissue of heart; Blood	NA	-	-	NA	-
134	Control	Male	60	AIH Recipient Before OLT	Tissue of native liver; Blood	NA	-	NA	NA	-
133	Control	Female	32	ACR After OLT	Tissue of donor liver; Blood	NA	-	-	NA	-
132	Control	Male	42	ACR After OLT	Tissue of donor liver; Blood	NA	-	-	NA	-
131	Control	Male	37	Normal recovery After OLT	Tissue of donor liver; Blood	NA	-	-	NA	-
130	Control	Male	52	HCC Recipient Before OLT	Tissue of native liver; Blood	NA	-	NA	NA	-
129	Control	Female	46	ACR After OLT	Tissue of donor liver; Blood	NA	-	-	NA	-
128	Control	Male	56	New hepatitis B After OLT	Tissue of donor liver; Blood	NA	-	-	NA	-
127	Control	Male	67	HCC Recipient Before OLT	Tissue of native liver; Blood	NA	-	NA	NA	-
126	Control	Male	57	HBV Cirrhosis Recipient Before OLT	Tissue of native liver; Blood	NA	-	NA	NA	-
125	Control	Male	36	HCC Recipient Before OLT	Tissue of native liver; Blood	NA	-	NA	NA	-
124	Control	Male	57	Infection After OLT	Tissue of skin; Blood	NA	-	-	NA	-

123	Control	Male	54	HBV Cirrhosis Recipient Before OLT	Tissue of native liver; Blood	NA	-	NA	NA	-
-----	---------	------	----	---------------------------------------	-------------------------------	----	---	----	----	---

122	Control	Male	22	PTLD After OLT	Tissue of donor liver/ileum; Blood	NA	-	-	NA	-
-----	---------	------	----	----------------	------------------------------------	----	---	---	----	---

NA: the method was not applied on detection of this patient sample

+: HTLV-1 infection was detected

-: HTLV-1 infection was not detected

Abbreviation

HCC: hepatocellular carcinoma

OLT: orthotopic liver transplantation

AR: acute rejection

IFI: invasive fungal infection

AIH: autoimmune hepatitis

Line 116-123: The DCD recipients were divided into two groups, the aGVHD and control groups. The aGVHD group comprised 7 patients diagnosed with aGVHD in different years (2015, 2017, 2018, and 2020). The control group consisted of 17 recipients, including post-transplant rejection patients (N=4), post-transplant infection patients (N=3), post-transplant regular recovery recipients (N=4), and pre-operative patients (N=6). For patients with aGVHD, post-transplant rejection, or post-transplant infection, samples were collected during the disease progression. Specimens were collected before surgery for preoperative control patients.

Line 128-131: For TargetSeq assay, fresh samples, including skin affected by a rash, peripheral blood, and biopsy tissues of livers, intestines, ventricular walls, and lungs, were harvested from 2 patients in the aGVHD group (ID 139 and ID141) and 17 recipients in the control group (Table S1).

Line 230-246: To reveal the cryptogenic infections associated with the development of aGVHD, TargetSeq was adopted to detect viral infection in samples from 2 patients (ID 139 and ID141) in the aGVHD group and 17 recipients in the control group (Fig. 1A). Detailed patient information and grouping principles were described in the Method and materials. In the aGVHD group, the peripheral blood, skins, or liver biopsy tissues from only 2 patients were acquired due to limits of sample collection (Fig. 2A). Likewise,

peripheral blood and disease-related lesion tissue specimens, including liver tissues, infectious site tissues, and lung metastases of liver tumors, were collected from the control group (Fig. 2A and Table S1). TargetSeq allowed us to determine the existence of viruses (HHV-6A, HTLV-1, HTLV-2, HTLV-4, EBV, adenovirus, CMV, and parvovirus-1), especially whether viruses were in the latent infection phase (Fraser et al., Science, 2014). Latent infection, where viral DNA had been integrated into the host genome, is a way for viruses to achieve persistent and lifelong infection in humans and participate in the pathogenesis of various complications (Bangham, Annu Rev Immunol, 2018). In the aGVHD and control groups, HTLV-2 and HTLV-4 were not detected and therefore excluded from subsequent analysis (Fig. 2A). HHV-6A and HTLV-1 were detected in all the aGVHD samples tested, while EBV and adenovirus were detected in some of the aGVHD samples (Fig. 2A).

Line 250-254: The presence of viruses was compared between groups by Fisher's exact test and corrected with Benjamini-Hochberg adjustment (Fig. 2C). Compared with the control group, the percentage of HTLV-1 positive patients was significantly higher in the aGVHD group (adjusted p-value = 0.0406) (Fig. 2C).

Line 257-264: Although HHV-6A was detected in all samples from the aGVHD group, HHV-6A was also present in the control group. There was no significant difference in the HHV-6A infection rates between the aGVHD and control groups (Fig. 2A and 2C). Therefore, these results preliminarily indicated the correlation between HTLV-1 infection and the occurrence of aGVHD. In addition, some of the viruses were not detected in the peripheral blood of the patients with tissue infections, which suggest it may be more reliable to perform virus screening on both tissue and peripheral blood (Fig. 2D).

Line 270-272: Notably, the recipient of the HTLV-1 positive donor was not infected due to the viral load of the donor is lower than the lowest load that can cause viral infection (9×10^4) (Sobata et al., Vox Sang, 2015).

Line 277-280: Given that, the possibility that the co-occurrences of the two rare diseases were a coincidence was less than 10^{-4} using a binomial probability calculation, suggesting that HTLV-1 infection should be closely related to aGVHD development after LT.

2. Fig. 3 about circulating TAX+CD68+ cells with features of APC is not clear or convincing.

Thanks for the comments. We apologize for the unclear statement in the original manuscript. To better address the reviewer's inquiry, we have added CD3, CD19, CD14, CD16, CD11c, CD123, and HLA-DR in the definition of seven Tax⁺CD68⁺ clusters for more precise classification (Table S5). As reported in the literature, cluster 29# (CD3⁻CD19⁻CD11c⁺CD123⁻HLA-DR⁺), 25# (CD3⁻CD19⁻CD14⁺CD11c⁺CD16⁺) and 26# (CD19⁻CD14⁻CD3⁺CD8⁺Tax⁺) were respectively cDCs, intermediate monocytes, and Tax⁺CD8⁺T cells (Iberg et al., 2017; Ziegler-Heitbrock et al., 2010).

Table S5. Characteristic phenotype of aGVHD group specific clusters.

Cluster	No.22	No.23	No.24	No.25	No.26	No.29	No.30	
Phenotype (+)				Tax				
				CD45				
				CD183				
				CD95				
				CD9				
				CD68				
		CD16	CD11c	CD194	CD11c	CD11c	CD11c	CD11c
		CD194	CD127	CD197	CD14	CD194	CD14	CD16
		CD21	CD16	CD38	CD127	CD3	CD194	CD194
		CD39	CD194	CD58	CD16	CD8	CD30	CD197
		CD38	CD38	CD69	CD194	CD38	CD38	CD21
			CD39	IgM	CD197	CD39	CD39	CD30
			CD58		CD21	HLA _DR	HLA _DR	CD39
			CD69		CD39	CD5	CD58	CD58
				CD58	IgM	CD69	IgM	
				IgM		CD8a		
Summary	nMo	ncMo	unknown	iMo	CD8 ⁺ T cell	cDC	nMo	

Abbreviation

iMo: intermediate monocyte

cDC: dendritic cell

nMo: non-classical monocyte

Our CyTOF panel did not include CD56 and CD66b to directly identify natural killer (NK) cells or neutrophils. Cluster 22#, 24#, 23#, and 30# were not classical monocytes, T cells, B cells, or DCs. Yet, since clusters 22#, 23#, and 30# have biomarker expressions as $CD3^-CD19^-CD14^+CD16^+HLA-DR^-CD68^+$, these cell clusters are classified as natural killer cells or monocytes. Because clusters 22#, 23#, and 30# expressed CD68 (a traditional marker of monocytes and macrophages) derived from peripheral blood, they are defined as non-classical monocytes. Cluster 24# does not express CD3, CD14, CD19, CD16, CD11c, or HLA-DR. We therefore do not see enough prevalent biomarkers to conclude Cluster 24# as NK cells or neutrophils when CD56 and CD66b were absent. Additionally, cluster 24# expresses activation markers such as CD38 (Malavasi et al., 2008) and CD69 (Ziegler et al., 1994), indicating cluster 24# was highly activated (**line 300-321**).

Original

Revised

Figure 3. Circulating $Tax^+ CD68^+$ immune cells are uncovered distinctively in aGVHD peripheral blood, mainly DCs and monocytes. (A) The two histograms visualize the

intensities of HTLV-1-specific Tax signal in CD45⁺ populations (Figure S1) from the two groups of PBMC samples. In aGVHD group, PBMCs from three aGVHD patients were collected at two-time points, T1 and T2, between onset and death. Ten patients in the control group received liver transplants and their PBMCs were collected at one postoperative time point when the complication occurred. PBMCs from aGVHD patients show high levels of HTLV-1 Tax expression. (B) t-SNE dimensional reduction was performed on CD45⁺ cells to explore the heterogeneity across patients. The t-SNE maps show their expression profiles. The color scale indicates the arcsinh-transformed signal intensity of Tax. The clusters with positive Tax expressions were marked through manual gating (red circles), primarily presented in the aGVHD group. (C) PhenoGraph and t-SNE depict a clustering result with 30 clusters identified automatically. The aGVHD group displays distinct cluster distributions from the control group. The Tax⁺ subpopulations were marked with hollow red circles (Table S5). (D) A heatmap depicts the median expression of cell markers in the seven Tax⁺ subpopulations. The HLA-DR⁺ subpopulations were marked with a hollow red rectangle. The color bars indicate arcsinh-transformed signal intensities of the proteins. (E) Percentages of the seven Tax⁺ subpopulations in CD45⁺ cells were compared between the aGVHD and control groups. Tax⁺ cells are presented mainly in the aGVHD group. (Intermediate monocyte: iMo; Non-classical monocyte: ncMo; Conventional dendritic cell: cDC; CD8⁺ T cell.)

Line 300-321: In Figure 3B, every dot represents a single cell, and color denotes the expression level of Tax protein. The spatial distributions of the immune cells were different between the aGVHD and control groups, suggesting different immune responses in these patients (Fig. 3B). We then clustered immune cells into 30 clusters based on the surface marker expressions with PhenoGraph(Chen et al., PLoS Comput Biol, 2016) (Fig. 3C). The major different clusters between the aGVHD and control groups were Tax⁺CD68⁺ clusters (Fig. 3B and S2). In consistent with the result of TargetSeq that HTLV-1 infection should be closely related to aGVHD development after LT, Tax⁺ cells were only detected in cells from the aGVHD group, which composed the clusters 22-26# and 29-30# (Fig. 3B and 3C). The cell phenotypes of the

Tax⁺ clusters are listed in Table S5. Cluster 23#, 25#, 29# and 30# were classified as CD3⁻CD19⁻CD11c⁺ CD123⁻ cells. Among them, Cluster 29# were HLA-DR⁺ cells, which can be defined as conventional dendritic cells (cDC cells). Dendritic cells (DCs), constituting the mononuclear phagocyte system (MPS), are considered to be the most potent APCs (Iberg et al., Trends in Immunology, 2017). Because CD68 is a selective marker for human monocytes and macrophages, Clusters 22#, 23# and 30# (CD3⁻CD19⁻CD14⁻CD16⁺HLA-DR⁻CD68⁺) were considered non-classical monocytes (Iqbal et al., Blood, 2014). Cluster 25# was intermediate monocytes (CD3⁻CD19⁻CD14⁺CD11c⁺CD16⁺CD68⁺). Cluster 24# (CD3⁻CD19⁻CD14⁻CD16⁻CD11c⁻CD123⁻HLA-DR⁻CD68⁺) expressed activation markers such as CD38 (Malavasi et al., Physiol Rev, 2008) and CD69 (Ziegler et al., Stem Cells, 1994), indicating these cells were highly activated. Meanwhile, cluster 26# (CD19⁻CD14⁻CD3⁺CD8⁺Tax⁺) was characterized as Tax⁺CD8⁺T cells (Fig. 3C, 3D and S2). Phenograph clustered 10 to 100 times more cells from the aGVHD group than the control group into the seven Tax⁺ clusters (Fig. 3E).

3. Fig. 4 IMC analysis of skin “triple-positive” CD4 T cells with expression of CD4, EIF1AY, and Tax protein is not clear or convincing. Single color and overlap need to be shown. In addition, human macrophages express CD4. To identify CD4⁺ T cells, CD3⁺CD4⁺ needs to be shown. Similarly, indications drawn from data in other figures are not convincing.

We are grateful for the reviewer’s comments and suggestions. To address the reviewer’s suggestion, the overlapping signals of CD3 and CD4 are presented in Figure S4 to identify CD4⁺ T cells. We have added single color images of all the 12 markers that were presented in Figures 4-6 with nuclear signals in Figures S4-9. We have revised the manuscript accordingly (**line 354-355, line 357-359, line 369-371, line 393-396, line 412-415**). The single Tax signal is shown in Figure 4 and magnified views of “Triple positive T cells” are added.

Line 354-355: The Tax⁺CD4⁺ cells also expressed CD3, so Tax⁺CD4⁺ cells were defined as CD4⁺ T cells (Fig. S4).

Line 357-359: Figure 4C and 4G demonstrate abundant C4d expression in immune cells that infiltrated in skin lesions and show co-expression of Tax and C4d protein.

Line 369-371: Furthermore, our results indicate that Tax expression was not detected in CD8⁺T cells or B cells in the skin (Fig. S5). Single signal of 12 lineage markers presented in Figure 4, 5 and S5 was presented in Figure S4, S6 and S7.

Line 393-396: The phenotypes and distribution of the four MNPs found by IMC in the skin lesions were presented by three representative patients with ID 139, ID140, and ID145 (Figure 5A-C). Five markers of MNPs from seven patients were shown in Figure S7 and S8.

Line 412-415: In the liver biopsies (liver graft) from female patients who received the organ from male donors (ID139), the Tax protein was detected in CD68⁺ EIF1AY⁺ cells, which confirmed donor-derived HTLV-1 infection from the liver graft (Fig. 6A, 4D and S9).

Figure 4. HTLV-1 specific Tax protein detected on the aGVHD skin tissues by IMC. Representative IMC images of the skin tissues from 7 aGVHD patients showed the overlap of Tax (red), DNA (blue) and different immune response markers. Displayed channels for each column were: (A) Tax (red)/DNA (blue); (B) CD4 (green)/Tax (red)/EIF1AY (white)/DNA (blue); (C) C4d (green)/ Tax (red)/ EIF1AY (white)/DNA (blue); (D) CD68 (green)/ Tax (red)/EIF1AY (white)/ DNA (blue). (E) Tax (red)/DNA (blue); (F) CD4 (green)/ Tax (red)/ DNA (blue); (G) C4d (green)/ Tax (red)/ DNA (blue); (H) CD68 (green)/ Tax (red)/ DNA (blue). Red arrows pinpointed the multi-signals overlap position and red box showed the magnified view of CD4⁺T cells with Tax signal. Scale bar =100 μ m for landscape view and scale bar =20 μ m for magnified view.

Figure S4. Single color images of “Triple positive T cells”. Representative IMC images of skin tissues from (A) 4 female aGVHD patients (ID 139, 141, 143 and 144) and (B) 3 male aGVHD patients (ID 140, 142, and 145). Displayed channels for each column were: DNA (blue), CD4 (green), Tax (red), CD3 (yellow), and EIF1AY (white). Scale bar =100 μ m.

Figure S5. CD4⁺ T cells detected on aGVHD skin tissues by IMC. Representative IMC images of skin tissues from seven aGVHD patients showed the overlap of DNA (blue) and CD3, CD4, and EIF1AY. Displayed channels for each column were: CD3 (green)/DNA (blue); CD4 (green)/DNA (blue); CD3 (green)/CD4 (red)/DNA (blue); EIF1AY (green)/DNA (blue). Red arrows pinpointed the CD3⁺CD4⁺T cells position, and red box showed the magnified view. Scale bar =100 μ m for landscape view and scale bar =20 μ m for magnified view.

Figure S6. Immune cells other than CD4⁺ T cells were unrelated with Tax⁺ in aGVHD patients. Representative IMC images of skin tissues from (A) four female and (B) three male aGVHD patients showed the overlap of Tax (red), DNA (blue), and different immunophenotypic markers. The same markers were shown in images in each column. Displayed channels for each column in (A) and (B) were: CD8 (green)/ Tax (red)/DNA (blue) and CD19 (green)/ Tax (red)/DNA (blue). Scale bar =100 μ m.

Figure S7. Single marker signals on aGVHD skin tissues by IMC. Representative IMC images of skin tissues from seven aGVHD patients showed the overlap of DNA (blue) and five different markers (green), including CD68, C4d, CD11c, HLA_DR, and CD123. Scale bar =100 µm.

Figure S8. Single marker signals on aGVHD skin tissues by IMC. Representative IMC images of skin tissues from seven aGVHD patients showed the overlap of DNA (blue) and four different markers (green), including CD8, CD19, CD16, and CD14. Scale bar =100 µm.

Figure S9. Single marker signals on liver graft, native liver, and bone marrow samples from aGVHD patients by IMC. Representative images of liver graft and native liver showed overlap of (A) DNA (blue)/CD68 (green); (B) DNA (blue)/Tax (green); (C) EIF1AY (green)/DNA (blue). Each row of figures displayed the same tissue type.

Reviewer 2

We truly appreciate the reviewers' gracious comments, especially the statistical reviews. These expertise comments indeed largely facilitate our revision and increase the readability of our manuscript. We would like to provide the following point-to-point response.

Overall, I think this is an interesting and well-written manuscript describing the association between HTLV-1 infection and acute graft vs. host disease following liver transplantation. There are a few minor grammatical errors that I assume were picked up by other reviewers. I am focusing on a statistical review, only.

1. I will note that "FFPE" is not defined.

We are grateful for the reviewer's suggestion. We have added definition of FFPE in manuscript (**line 138-140**).

Line 138-140: The specimens for IMC detection were formalin fixation and paraffin embedding (FFPE) sections of native livers, skin lesions, liver biopsies (graft tissues), and bone marrows obtained from 7 aGVHD patients in the aGVHD group (Table S1).

2. In the Statistical Analysis section, the authors only report performing Fisher's exact tests to compare Group A to Group B and to Group C. In the Results section, regarding detection of HTLV-1 infection: Was RT-PCR used in Groups A-D to determine infection with HTLV-1? My assumption based on the reading of the methods and results is that HTLV-1 infection was determined by the methods described in the results and not using RT-PCR.

Thanks a lot for the inquiry. We apologize for the unclear statement in the original manuscript. Samples from 2 group A patients and 17 group B-C patients were assessed with hybridization probes (TargetSeq) and signals of HTLV-1 were captured by RT-PCR and HTLV-1 infection were detected in group A patients. HTLV-1 infection was then determined by an overlapping testing method, namely imaging mass cytometry (IMC), in skin sections from the 7 aGVHD patients, including group A and group D

patients. For better clarity, we have listed the patients from which the samples were tested, the corresponding detection methods used, and the detection results of HTLV-1 infection in the table below.

ID	Revised Group	Original Group	Sample type	HTLV-1 detection method				HTLV-1 detection summary
				IMC	Targetseq	CyTOF	ELISA	
145	GVHD	D	FFPE of skin, native liver; Blood	+	NA	NA	+	+
144	GVHD	D	FFPE of skin, native liver; Blood	+	NA	NA	+	+
143	GVHD	D	FFPE of skin	+	NA	NA	NA	+
142	GVHD	D	FFPE of skin	+	NA	NA	NA	+
141	GVHD	A	Tissue/FFPE of skin, native/donor liver; Blood	+	+	+	NA	+
140	GVHD	A	Tissue/FFPE of skin, native/donor liver; Blood	+	NA	+	NA	+
139	GVHD	A	Tissue/FFPE of skin, native/donor liver; Blood	+	+	+	NA	+
138	Control	B	Tissue of donor liver; Blood	NA	-	-	NA	-
137	Control	B	Tissue of donor liver; Blood	NA	-	-	NA	-
136	Control	B	Tissue of lung; Blood	NA	-	-	NA	-
135	Control	B	Tissue of heart; Blood	NA	-	-	NA	-
134	Control	C	Tissue of native liver; Blood	NA	-	NA	NA	-
133	Control	B	Tissue of donor liver; Blood	NA	-	-	NA	-
132	Control	B	Tissue of donor liver; Blood	NA	-	-	NA	-
131	Control	B	Tissue of donor liver; Blood	NA	-	-	NA	-
130	Control	C	Tissue of native liver; Blood	NA	-	NA	NA	-
129	Control	B	Tissue of donor liver; Blood	NA	-	-	NA	-
128	Control	B	Tissue of donor liver; Blood	NA	-	-	NA	-
127	Control	C	Tissue of native liver; Blood	NA	-	NA	NA	-
126	Control	C	Tissue of native liver; Blood	NA	-	NA	NA	-
125	Control	C	Tissue of native liver; Blood	NA	-	NA	NA	-
124	Control	B	Tissue of skin; Blood	NA	-	-	NA	-
123	Control	C	Tissue of native liver; Blood	NA	-	NA	NA	-
122	Control	B	Tissue of donor liver/ileum; Blood	NA	-	-	NA	-

NA: the method was not applied on detection of this patient sample

+: HTLV-1 infection was detected

-: HTLV-1 infection was not detected

Figure 1. A schematic of the experimental design. (A) Multiple viral DNA and RNA sequences were detected in 37 specimens from the patients in the aGVHD group (N=2) and the control group (N=17). Tested specimens included skins with rash, peripheral blood, liver grafts, intestines, ventricular walls, and lungs. (B) CyTOF was performed on a total of 16 samples of peripheral blood mononuclear cells (PBMCs) from the patients in the aGVHD group (N=3) and the control group (N=10). (C) 17 FFPEs from the patients in the aGVHD group (N=7), including 7 skin tissues, 5 native liver tissues, 3 graft liver tissues, and 2 bone marrow tissues, were assessed with IMC.

Samples from the 24 recipients were respectively assessed by TargetSeq, mass cytometry by time of flight (CyTOF), ELISA, and imaging mass cytometry (IMC). After simplifying grouping, samples from 2 patients in the aGVHD group and 17 patients in the control group were assessed with TargetSeq. CyTOF was applied on the PBMCs from 3 aGVHD group patients and 10 control group patients. ELISA was also performed on blood serums from 2 patients in the aGVHD group. IMC was the multiplex protein overlapping test method that was applied on skin tissues from 7 patients in the aGVHD group. The HTLV-1 infection results of four methods are summarized in the Table S1 which is a detailed version of the table above. Therefore, HTLV-1 infection in the aGVHD group was independently determined by four detection methods and IMC was an overlapping testing method to detect HTLV-1 infection on samples from 7 aGVHD patients.

3. Group A (the group with aGVHD) had a sample size of 3 but one of the samples (ID #140) suffered from degradation during transport and could not be assessed. HTLV-1 was detected in samples from the other two (ID #s 139 and 141). Therefore, when comparing Group A to Group B (with no HTLV-1 infections), a Fisher's exact test of 2/2 (100%) vs. 0/11 (0%) was performed, resulting in a p-value of 0.0128. I replicated this. Comparing Group A to Group C (with n HTLV-1 infections), a Fisher's exact test of 2/2 (100%) vs. 0/6 (0%) was performed resulting in a p-value of 0.0357, also replicated by me. While these probabilities are below the nominal level of 0.05, a sample size of 2 in Group A is not compelling. Additionally, a sample size of 2 is extremely unlikely to be representative of the target population. Further, there was no adjustment made for multiple comparisons. A simple and acceptable multiple comparison adjustment for this case could be a Bonferroni correction for the number of comparisons being made, in this case 2. Therefore, a Bonferroni-adjusted type I error rate for p-value comparison would be $0.05/2=0.025$. If this were used, then Group A vs. Group B would still result in a "significant" difference while Group A vs. Group C would not. What would then be the conclusion?

We are grateful for the reviewer's expertise comments and inquiry. If the Bonferroni correction was performed, there is no significant difference between group A and group C with only two aGVHD patients. However, we would still claim that our final conclusions should still be trustworthy because HTLV-1 infection was detected by imaging mass cytometry (IMC) in all 7 aGVHD patients. Although only samples from 2 aGHVD patients were assessed with TargetSeq, HTLV-1 signals were positive in skin sections from 7 aGVHD patients assessed by IMC. Sample size of the aGVHD patients was expanded to 7 by application of IMC to eliminate the contingency caused by the sample size of 2. Comparing the HTLV-1 infection rates between the aGVHD and control groups, a Fisher's exact test of 7/7 (100%) vs. 0/17 (0%) is performed, resulting in a p-value of 0.00001. Therefore, the HTLV-1 infection was statistically correlated with the occurrence of aGHVD.

Meanwhile, as indicated in the response of the 2nd question, we simplify the grouping in the revised manuscript. The statistical analysis is therefore re-performed accordingly (**line 230-246** and **line 250-254**). TargetSeq was performed to detect eight non-hepatotropic viruses on samples from 2 aGVHD group patients and 17 control group patients. Infection HTLV-2 and HTLV-4 were not detected in the aGVHD or control groups by TargetSeq, so they are excluded from subsequent analysis. The Fisher's exact test is performed to compare the infection patient counts of the remaining six viruses between the aGVHD and control groups. When comparing the aGVHD to control group (HTLV-1 infection), a Fisher's exact test of 2/2 (100%) vs. 0/17 (0%) is performed, resulting in a p-value of 0.0058. Since Fisher's exact tests are performed 6 times, p-values are corrected p-values with Benjamini-Hochberg adjustment, using the formula $q\text{-value} = p\text{-value} \times \frac{n}{k}$, $n =$ numbers of Fisher's exact test ($k =$ the individual p-value's rank, $q\text{-value} =$ adjusted p-value). Compared with the control group, the percentage of HTLV-1 positive patients is significantly higher in the aGVHD group with a q-value of 0.0406 ($0.0056 \times \frac{6}{1}$) (**line 230-246** and **line 250-254**).

Line 230-246: To reveal the cryptogenic infections associated with the development of aGVHD, TargetSeq was adopted to detect viral infection in samples from 2 patients (ID 139 and ID141) in the aGVHD group and 17 recipients in the control group (Fig. 1A). Detailed patient information and grouping principles were described in the Method and materials. In the aGVHD group, the peripheral blood, skins, or liver biopsy tissues from only 2 patients were acquired due to limits of sample collection (Fig. 2A). Likewise, peripheral blood and disease-related lesion tissue specimens, including liver tissues, infectious site tissues, and lung metastases of liver tumors, were collected from the control group (Fig. 2A and Table S1). TargetSeq allowed us to determine the existence of viruses (HHV-6A, HTLV-1, HTLV-2, HTLV-4, EBV, adenovirus, CMV, and parvovirus-1), especially whether viruses were in the latent infection phase (Fraser et al., Science, 2014). Latent infection, where viral DNA had been integrated into the host genome, is a way for viruses to achieve persistent and lifelong infection in humans and

participate in the pathogenesis of various complications (Bangham, *Annu Rev Immunol*, 2018). In the aGVHD and control groups, HTLV-2 and HTLV-4 were not detected and therefore excluded from subsequent analysis (Fig. 2A). HHV-6A and HTLV-1 were detected in all the aGVHD samples tested, while EBV and adenovirus were detected in some of the aGVHD samples (Fig. 2A).

Line 250-254: The presence of viruses was compared between groups by Fisher's exact test and corrected with Benjamini-Hochberg adjustment (Fig. 2C). Compared with the control group, the percentage of HTLV-1 positive patients was significantly higher in the aGVHD group (adjusted p-value = 0.0406) (Fig. 2C).

Figure 2. Multiple viral cDNA probe detections indicate that HTLV-1 presents exclusively in all the peripheral blood and tissues of aGVHD patients. (B) Patient numbers of virus infection and host integration were counted. (C) The percentages of virus infection rates in each group were calculated and compared between the aGVHD and control groups. HTLV-1 was detected in all the 6 samples from the 2 patients in the aGVHD group, but none of the samples from the control group. All p-values were calculated using Fisher's exact test and corrected with Benjamini-Hochberg adjustment.

***p < 0.05. (D) The percentage of patients with virus detected in either peripheral blood or tissue samples.**

4. At the end of the first paragraph of the Results section, the authors state, “Given the overall positive rate of HTLV-1 in all tested samples was 6/37, the possibility (p-value) that it was a coincidence was less than 10^{-7} , which was practically preclusive.” Please delete the last phrase (“which was practically preclusive”) and note this language was used again in the first paragraph of the discussion. The word “preclusive” is not being used in an appropriate way and is not typical language used for interpreting p-values. I spent a long time trying to understand where these numbers came from (deduced from Figure 2A). Please refer the reader to Figure 2 in this text or better describe where the numbers 6 and 37 come from. The calculation of 10^{-7} is assumed to be a binomial probability where the sample size is 37, the number positive is 6, and the probability of 1 sample being positive is 0.005 (from the prevalence of HTLV-1 in the cohort of 400 patients). I don’t know what criteria is used to formally diagnose a patient with HTLV-1—is it a positive from any of possibly multiple tissue/blood samples in Groups A-D (assuming not by PCR)? At any rate, the estimated prevalence rate for HTLV-1 of 0.005 is the probability of a person being diagnosed with HTLV-1, not the probability of a single tissue or blood sample being positive for HTLV-1, which I assume would be higher since multiple tissues from the same person could yield positive results. It is not appropriate to use this value in your binomial probability calculation unless only a single tissue positive for HTLV-1 is equivalent to or highly correlated with a positive RT-PCR test (which was used for the cohort of 400 to estimate the prevalence of 0.005). Additionally, your statistical analysis section should describe using a binomial probability calculation for this assessment—I had to guess what you did and ultimately calculated a probability close to what you reported.

We are grateful for the reviewer’s professional suggestion and inquiry in statistical analysis. Per the reviewer’s comment, we have removed the inaccurate statement (“which was practically preclusive”) from our manuscript. The calculation of 10^{-7} used binomial probability calculation ($(\frac{6}{37})^6 \times (\frac{31}{37})^{31} \approx 7.5 \times 10^{-8}$). In the original

manuscript, 37 and 6 are respectively the number of all samples and the number of HTLV-1 positive samples collected from patients in both aGVHD (n = 2) and control (n = 17) group. Specifically, peripheral blood, liver, and skin tissues were separately collected from 2 patients in the aGVHD group and therefore a total of 6 samples were detected for HTLV-1 infection by TargetSeq. To avoid yielding artificially positive results, we have modified the manuscript to replace the sample infection rate with the patient infection rate and re-performed the statistical analysis (**line 250-254**). According to the cohort of 400 patients, the probability of 1 patient being HTLV-1 in our center is 0.005. Herein, 2/19 patients were detected to be HTLV-1 positive by TargetSeq. Therefore, the possibility (p-value) of detecting HTLV-1 in all aGVHD patients but no control patients as coincidental is less than 10^{-4} using a binomial probability calculation ($0.005^2 \times (1 - 0.005)^{19-2} \approx 2.3 \times 10^{-5}$) (**line 278-280**).

Line 250-254: The presence of viruses was compared between groups by Fisher's exact test and corrected with Benjamini-Hochberg adjustment (Fig. 2C). Compared with the control group, the percentage of HTLV-1 positive patients was significantly higher in the aGVHD group (adjusted p-value = 0.0406) (Fig. 2C).

Line 278-280: Given that, the possibility that the co-occurrences of the two rare diseases were a coincidence was less than 10^{-4} using a binomial probability calculation, suggesting that HTLV-1 infection should be closely related to aGVHD development after LT.

5. Describe in the Statistical Analysis section how you calculated the 95% CI for the prevalence of HTLV-1+ reported in the second paragraph of the Results. I cannot replicate this using standard methods.

Thanks a lot for the inquiry. Here, the total number of tested patients was 400, and 2 were detected positive. Therefore, the goal was to estimate the possibility of a binomial distribution and its confidence interval. We adopted two methods for confidence interval estimation, namely the Wilson method and the Agresti-Coull method. Because the coverage of the Agresti-Coull confidence interval of [0.01%, 1.93%] is larger than

the Wilson confidence interval of [0.14%, 1.8%], we used the Agresti-Coull confidence interval in the manuscript, which is

$$\left[\tilde{p} - a \sqrt{\frac{\tilde{p}(1 - \tilde{p})}{\tilde{n}}}, \tilde{p} + a \sqrt{\frac{\tilde{p}(1 - \tilde{p})}{\tilde{n}}} \right],$$

in which

$$a = z_{1-\frac{\alpha}{2}}, \tilde{n} = n + a^2, \tilde{p} = \frac{1}{\tilde{n}} \left(\sum_{i=1}^n X_i + \frac{1}{2} a^2 \right).$$

6. In the Results subsection “Triple-positive T cells...”, second half of the first paragraph, Group D results are presented for the first time with results only by ELISA (but no statistics to support conclusion). This section then refers the reader to Table 1 and reports p-values from the comparison of Groups A and B, this time including all 3 samples from Group A, but it was already established that Group A ID #140 was not viable. Unless I misunderstand the type of testing performed on the peripheral blood and liver graft samples in Group A, you already presented these results with different sample sizes (2 vs. 3). I believe this sentence should be deleted and, instead, refer the reader back to the results presented in “HTLV-1 only found in aGVHD patients...” if necessary. The final sentence of this first paragraph concludes that the four methods cross-validated that the 7 aGVHD patients (only) were infected with HTLV-1. This is a bit of a strong statement given that testing via TargetSeq in Group A consisted of only n=2 and only ELISA was performed in n=4 in Group D. Your results would be stronger if there was an overlapping testing mechanism in Groups A and D.

Thanks a lot for the comments and suggestions. We apologize for the unclear statement in the original manuscript. Imaging mass cytometry (IMC) was the multiplex protein overlapping test method which was applied to all skin lesion samples from the 7 aGVHD patients. The HTLV-1 specific Tax protein was detected. We have listed detection methods applied on 24 recipients, HTLV-1 infection summary, and simplified grouping in Table S1.

Table S1 Patients' information profile.

ID	Group	Sex	Age	Diagnosis	Sample type	HTLV-1 detection method				HTLV-1 detection summary
						IMC	Targetseq	CyTOF	ELISA	
145	GVHD	Male	41	aGVHD After OLT	FFPE of skin, native liver; Blood	+	NA	NA	+	+
144	GVHD	Female	62	aGVHD After OLT	FFPE of skin, native liver; Blood	+	NA	NA	+	+
143	GVHD	Female	47	aGVHD After OLT	FFPE of skin	+	NA	NA	NA	+
142	GVHD	Male	50	aGVHD After OLT	FFPE of skin	+	NA	NA	NA	+
141	GVHD	Female	54	aGVHD After OLT	Tissue/FFPE of skin, native/donor liver; Blood	+	+	+	NA	+
140	GVHD	Male	45	aGVHD After OLT	Tissue/FFPE of skin, native/donor liver; Blood	+	NA	+	NA	+
139	GVHD	Female	63	aGVHD After OLT	Tissue/FFPE of skin, native/donor liver; Blood	+	+	+	NA	+
138	Control	Male	48	Normal recovery After OLT	Tissue of donor liver; Blood	NA	-	-	NA	-
137	Control	Male	59	ACR After OLT	Tissue of donor liver; Blood	NA	-	-	NA	-
136	Control	Male	23	HCC recurrence After OLT	Tissue of lung; Blood	NA	-	-	NA	-
135	Control	Female	31	IFI After OLT	Tissue of heart; Blood	NA	-	-	NA	-
134	Control	Male	60	AIH Recipient Before OLT	Tissue of native liver; Blood	NA	-	NA	NA	-
133	Control	Female	32	ACR After OLT	Tissue of donor liver; Blood	NA	-	-	NA	-
132	Control	Male	42	ACR After OLT	Tissue of donor liver; Blood	NA	-	-	NA	-
131	Control	Male	37	Normal recovery After OLT	Tissue of donor liver; Blood	NA	-	-	NA	-
130	Control	Male	52	HCC Recipient Before OLT	Tissue of native liver; Blood	NA	-	NA	NA	-
129	Control	Female	46	ACR After OLT	Tissue of donor liver; Blood	NA	-	-	NA	-

128	Control	Male	56	New hepatitis B After OLT	Tissue of donor liver; Blood	NA	-	-	NA	-
127	Control	Male	67	HCC Recipient Before OLT	Tissue of native liver; Blood	NA	-	NA	NA	-
126	Control	Male	57	HBV Cirrhosis Recipient Before OLT	Tissue of native liver; Blood	NA	-	NA	NA	-
125	Control	Male	36	HCC Recipient Before OLT	Tissue of native liver; Blood	NA	-	NA	NA	-
124	Control	Male	57	Infection After OLT	Tissue of skin; Blood	NA	-	-	NA	-
123	Control	Male	54	HBV Cirrhosis Recipient Before OLT	Tissue of native liver; Blood	NA	-	NA	NA	-
122	Control	Male	22	PTLD After OLT	Tissue of donor liver/ileum; Blood	NA	-	-	NA	-

NA: the method was not applied on detection of this patient sample

+: HTLV-1 infection was detected

-: HTLV-1 infection was not detected

Abbreviation

HCC: hepatocellular carcinoma

OLT: orthotopic liver transplantation

AR: acute rejection

IFI: invasive fungal infection

AIH: autoimmune hepatitis

The blood and tissue samples from 2 aGVHD group patients (ID 139 and 141) and 17 control group patients were assessed with TargetSeq. HTLV-1 infection was only detected in 2 aGVHD group patients. Tax protein, an HTLV-1 specific RNA transcription enhancer, was captured by Tax antibody tagged by metal isotope in the IMC and CyTOF detection. CyTOF was applied on PBMCs from 3 aGVHD group patients (ID 139-141) and 10 control group patients. IMC was conducted to test all skin tissues from 7 aGVHD group patients. HTLV-1 signals were detected by Tax antibody in the PBMCs from aGVHD patients (ID 139-141) and skin sections from 7 aGVHD

patients as assessed with IMC (ID 139-145). Positive Tax signals on skin tissues of 7 aGVHD patients were pinpointed by red arrows, which confirmed that 7 aGVHD patients were infected with HTLV-1 (Figure 4A and 4E). Additionally, ELISA was performed on blood serums from patients ID144 and ID145 and high titers of HTLV-1-specific antigen were detected.

Figure 4. HTLV-1 specific Tax protein detected on the aGVHD skin tissues by IMC. Representative IMC images of the skin tissues from 7 aGVHD patients showed the overlap of Tax (red), DNA (blue) and different immune response markers. Displayed channels for each column were: (A) Tax (red)/DNA (blue); (B) CD4 (green)/Tax (red)/EIF1AY (white)/DNA (blue); (C) C4d (green)/ Tax (red)/ EIF1AY (white)/DNA (blue); (D) CD68 (green)/ Tax (red)/ EIF1AY (white)/DNA (blue);

(D) CD68 (green)/ Tax (red)/EIF1AY (white)/ DNA (blue). (E) Tax (red)/DNA (blue); (F) CD4 (green)/ Tax (red)/ DNA (blue); (G) C4d (green)/ Tax (red)/ DNA (blue); (H) CD68 (green)/ Tax (red)/ DNA (blue). Red arrows pinpointed the multi-signals overlap position and red box showed the magnified view of CD4⁺T cells with Tax signal. Scale bar =100 μ m for landscape view and scale bar =20 μ m for magnified view.

7. In the Discussion, first paragraph, you mention “According to probability analysis, it is statistically preclusive that HTLV-1 infection and aGVHD accidentally occurred at the same time for all seven aGVHD patients.” However, I don’t see analysis results presented where all 7 were combined with a probability calculated like you did for Group A alone. There are no strengths or weaknesses presented.

Thanks a lot for the inquiry. In original manuscript, we described the comparison between the aGVHD and control group at the end of the first paragraph of the third section in the Result. Since we have simplified the grouping in the revised manuscript, we therefore have re-performed statistical analysis (**line 370-378**). Fisher’s exact test is adopted to test the differences in infected cases of six viruses between the control and aGVHD groups, and p-values are corrected with Benjamini-Hochberg adjustment. When comparing the aGVHD to control group (HTLV-1 infection), a Fisher’s exact test of 7/7 (100%) vs. 0/17 (0%) results in a p-value of 0.00001. Since Fisher's exact tests are performed 6 times, the percentage of HTLV-1 positive patients is significantly higher in the aGVHD group with an adjusted p-value of 0.00006 ($0.00001 \times \frac{6}{1}$) compared with the control group. HTLV-1 virus was detected in all 7 aGVHD patients by CyTOF, ELISA, TargetSeq, and IMC. HTLV-1 infected T cells were found in skin lesions of 7 aGVHD patients by IMC. Samples fixed in paraformaldehyde from 5 aGVHD patients (ID 140, 142-145) could not be assessed with TargetSeq, so their detection results of HHV-6A, HTLV-2, HTLV-4, EBV, adenovirus, CMV, and parvovirus-1 were not available. HTLV-2 and HTLV-4 were not detected by TargetSeq and therefore were excluded from subsequent analysis. We found their past infection of these five viruses from clinical records and summarized detection results of six virus

in our study, which is listed in Table 1 for statistical analysis. Therefore, it was demonstrated that the occurrence of GVHD is inseparable from HTLV-1 infection based on statistical analysis.

Line 370-378: To fully display the HTLV-1 detection result assessed with multiple methods in the discovery cohort, the positive rate and specific detection information of HTLV-1 were listed in Table 1 and Table S1. By tracking the clinical records, we integrated the PCR detection results of HHV-6A, EBV, adenovirus, CMV, and parvovirus-1 in all the 24 patients in the discovery cohort and performed statistical analysis between the two groups (Table 1). By Fisher's exact test and corrected with Benjamini-Hochberg adjustment, significant differences in positive rate of HTLV-1 detection were found between the control and aGVHD groups (adjusted p-value =0.00006). The four methods cross-validated that all and only the 7 aGVHD patients were infected with HTLV-1.

Table 1. Summary of virus detection results and joint analysis.

Virus	Number of cases (Infected/Total)	
	Control Group	aGVHD Group
HHV-6A	9/17	7/7
HTLV-1	0/17	7/7***
EBV	10/17	3/7
ADENOVIRUS	1/17	1/7
CMV	11/17	2/7
PARVOVIRUS-1	3/17	0/7

*# The number of infected cases of the aGVHD group were compared with the control group by Fisher's exact test and corrected p-values by Benjamini-Hochberg adjustment. (***) $p < 0.001$*

8. The very small sample sizes are a limiting factor of these results. More emphasis should be placed on the need for larger studies adequately powered to detect the relationship between HTLV-1 and aGVHD.

Thanks a lot for the suggestion. Although the discovery cohort of 7 aGVHD patients is not that sizable, our study still revealed a high correlation between the pathogenesis of aGVHD and HTLV-1 infection. We are conducting larger studies to explore the relationship between HTLV-1 and aGVHD, so we have added an outlook for future research in the text per the reviewer's suggestion (**line 458-461**).

Line 458-461: *In this study, the specificity of HTLV-1 virus infection in aGVHD patients after LT was preliminarily revealed, so a more extensive multicenter study to examine the relationship between HTLV-1 and aGVHD will be beneficial.*

9. Table 1 is problematic in that it implies that all three samples in Group A were used for the calculation of reported statistics when really only 2 were used (for TargetSeq). The p-values reported in the footnote are for 2/2 in Group A not 3/3. Consider changing Method to "TargetSeq or IMC" but then add a footnote that the p-values were calculated only from TargetSeq analysis. I don't see enough description in the text of the IMC results to understand how HTLV-1+ was determined via that method.

Thanks a lot for the comments. We have revised table 1 for better clarity. Imaging mass cytometry (IMC), an approach that combines mass cytometry by time of flight (CyTOF) with immunocytochemistry (ICC) and immunohistochemistry (IHC) techniques, can detect over 20 protein expression levels and modifications simultaneously at a cellular resolution of 1 μm on one tissue specimen (Figure 1C). Tax protein, an HTLV-1 specific RNA transcription enhancer, was captured by Tax antibody tagged by metal isotope and detected by IMC and CyTOF. Per the reviewer's request, we have introduced detailed information on Tax protein in CyTOF results and two panels of

antibodies in CyTOF and IMC detection, respectively (**line 290-293** and **line 331-338**). Therefore, HTLV-1 positive was presented as Tax⁺ signals on skin tissues of 7 aGVHD patients and pinpointed by red arrows (Figure 4A and 4E). Table 1 listed the all the HTLV-1 detection efforts by multiple methods in the original manuscript. We have revised table 1 and integrated infections of the six viruses in the discovery cohort.

Line 290-293: The panel contained 30 lineage markers to distinguish the significant leukocyte subsets and a marker to trace HTLV-1 infection at the cellular level. Mainly, a metal-labeled antibody against Tax protein, an HTLV-1 specific RNA transcription enhancer, (Boxus et al., *Retrovirology*, 2008; Brauweiler et al., *Virology*, 1997) was included in the panel.

Line 331-338: An IMC panel of 19 metal isotope-tagged antibodies was designed to acquire a global overview of the immune cells in the skin lesions and reveal the interactions between HTLV-1 and immune cells (Table S4). The panel contained 2 structure protein markers, 13 immune cell markers, an HTLV-1 specific marker (Tax protein), and a Y chromosome marker to track the source of cells. IMC was performed on skin lesions from all the 7 patients in the aGVHD group. Among them, 4 female patients (ID139, ID141, ID143, and ID144) received livers from male donors.

Figure 1. A schematic of the experimental design. (A) Multiple viral DNA and RNA sequences were detected in 37 specimens from the patients in the aGVHD group (N=2) and the control group (N=17). Tested specimens included skins with rash, peripheral blood, liver grafts, intestines, ventricular walls, and lungs. (B) CyTOF was performed on a total of 16 samples of peripheral blood mononuclear cells (PBMCs) from the patients in the aGVHD group (N=3) and the control group (N=10). (C) 17 FFPEs from the patients in the aGVHD group (N=7), including 7 skin tissues, 5 native liver tissues, 3 graft liver tissues, and 2 bone marrow tissues, were assessed with IMC.

Figure 4. HTLV-1 specific Tax protein detected on the aGVHD skin tissues by IMC. Representative IMC images of the skin tissues from 7 aGVHD patients showed the overlap of Tax (red), DNA (blue) and different immune response markers. Displayed channels for each column were: (A) Tax (red)/DNA (blue); (B) CD4 (green)/Tax (red)/ EIF1AY (white)/DNA (blue); (C) C4d (green)/ Tax (red)/ EIF1AY

(white)/DNA (blue); (D) CD68 (green)/ Tax (red)/EIF1AY (white)/ DNA (blue). (E) Tax (red)/DNA (blue); (F) CD4 (green)/ Tax (red)/ DNA (blue); (G) C4d (green)/ Tax (red)/ DNA (blue); (H) CD68 (green)/ Tax (red)/ DNA (blue). Red arrows pinpointed the multi-signals overlap position and red box showed the magnified view of CD4⁺T cells with Tax signal. Scale bar =100 μ m for landscape view and scale bar =20 μ m for magnified view.

10. I note that Figure 2A reports that, in particular, Groups B2 and B3 use tissue samples not obtained in the other groups. Is it typical to compare different types of tissue samples?

Thanks a lot for the inquiry. Yes, it is typical to compare different types of tissue samples (Cosio et al., Am J Transplant, 2004; Ozoya et al., Infect Disord Drug Targets, 2016). In our study, HTLV-1, HHV-6A, EBV, and adenovirus infections were found in the peripheral blood or skin lesions of 2 patients with aGVHD, indicating viral infection may be related to the occurrence of aGVHD. Therefore, we need to compare the relationship between different post-transplant complications and viral infection. Typically, there is post-transplant lymphoproliferative disease (PTLD), which is generally considered to be highly associated with EBV infection. Due to the different sites of EBV infection, PTLD lesions can occur in the throat, ileocecal, and mesenteric lymph nodes. The clinical manifestations are diverse, including dyspnea, abdominal mass, intestinal perforation. In addition, the post-transplant infectious diseases involve more body sections, including the respiratory tract, solid organs, the urinary system, and the nervous system. In our study, we focus the types of pathogenic microorganisms in the infected tissue and their roles in the disease, so the types of tissue samples were not the priority.

11. Figures 2B and 2D: The y-axis in both figures should be either “Sample Counts” or “Sample Proportions”, but not both. Update appropriately.

We are grateful for the reviewer’s suggestion. We have modified Figure 2B and 2D to address the reviewer’s request and revised the manuscript accordingly (**line 260-263**).

Figure 2. Multiple viral cDNA probe detections indicate that HTLV-1 presents exclusively in all the peripheral blood and tissues of aGVHD patients. (A) The heatmap depicts the detection of virus or virus/host integration in each sample from 2 patients in the aGVHD group and 17 patients in the control group. Negative, positive, and host integration results are marked as blue, yellow, and red, respectively. (B) Patient numbers of virus infection and host integration were counted. (C) The percentages of virus infection rates in each group were calculated and compared between the aGVHD and control groups. HTLV-1 was detected in all the 6 samples from the 2 patients in the aGVHD group, but none of the samples from the control group. All p-values were calculated using Fisher's exact test and corrected with Benjamini-Hochberg adjustment. * $p < 0.05$. (D) The percentage of patients with virus detected in either peripheral blood or tissue samples. (E) The heatmap depicts basic clinical information of the 400 patients in the validation cohort, which included age, body mass index (BMI), sample collection time, preoperative diagnosis, and concomitant diseases. Red arrows with annotations of the viral loads marked two HTLV-1 infection patients. (Benign ESLD: benign end-stage liver disease, Malignant LD: malignant liver disease, Complications after LT: complications after liver transplantation, NC: no concomitant disease.)

Line 260-263: In addition, the virus was undetectable in the peripheral blood of more than half of the virus-infected patients, so it may be more reliable to perform virus screening on both tissue and peripheral blood (Fig. 2D).

12. Table S1: The second column is unclear. The header (“HTLV(+/-) 1.Targetseq TM / 2. Cytof / 3. IMC”) does not sufficiently explain the content of the cells in that column (e.g., “C (skin, native liver) / HTLV(+) / 3 B/HTLV-1(+) (ELISA)”).

Thanks a lot for the comments. We apologize for the unclear statement in the original Table S. We have revised Table S1 to enhance clarity. The column of sample type indicates all sample types of each patient used in our research. The sub-column of the HTLV-1 detection method displays the methods used to detect the HTLV-1 infection in each patient in our study. For example, Targetseq, CyTOF, and IMC were performed to detect HTLV-1 infection in the patient with ID 141 and the plus sign indicates the detection result was positive.

Table S1 Patients’ information profile.

ID	Group	Sex	Age	Diagnosis	Sample type	HTLV-1 detection method				HTLV-1 detection summary
						IMC	Targetseq	CyTOF	ELISA	
145	GVHD	Male	41	aGVHD After OLT	FFPE of skin, native liver; Blood	+	NA	NA	+	+
144	GVHD	Female	62	aGVHD After OLT	FFPE of skin, native liver; Blood	+	NA	NA	+	+
143	GVHD	Female	47	aGVHD After OLT	FFPE of skin	+	NA	NA	NA	+
142	GVHD	Male	50	aGVHD After OLT	FFPE of skin	+	NA	NA	NA	+
141	GVHD	Female	54	aGVHD After OLT	Tissue/FFPE of skin, native/donor liver; Blood	+	+	+	NA	+
140	GVHD	Male	45	aGVHD After OLT	Tissue/FFPE of skin, native/donor liver; Blood	+	NA	+	NA	+
139	GVHD	Female	63	aGVHD After OLT	Tissue/FFPE of skin, native/donor liver; Blood	+	+	+	NA	+
138	Control	Male	48	Normal recovery After OLT	Tissue of donor liver; Blood	NA	-	-	NA	-
137	Control	Male	59	ACR After OLT	Tissue of donor liver; Blood	NA	-	-	NA	-

136	Control	Male	23	HCC recurrence After OLT	Tissue of lung; Blood	NA	-	-	NA	-
135	Control	Female	31	IFI After OLT	Tissue of heart; Blood	NA	-	-	NA	-
134	Control	Male	60	AIH Recipient Before OLT	Tissue of native liver; Blood	NA	-	NA	NA	-
133	Control	Female	32	ACR After OLT	Tissue of donor liver; Blood	NA	-	-	NA	-
132	Control	Male	42	ACR After OLT	Tissue of donor liver; Blood	NA	-	-	NA	-
131	Control	Male	37	Normal recovery After OLT	Tissue of donor liver; Blood	NA	-	-	NA	-
130	Control	Male	52	HCC Recipient Before OLT	Tissue of native liver; Blood	NA	-	NA	NA	-
129	Control	Female	46	ACR After OLT	Tissue of donor liver; Blood	NA	-	-	NA	-
128	Control	Male	56	New hepatitis B After OLT	Tissue of donor liver; Blood	NA	-	-	NA	-
127	Control	Male	67	HCC Recipient Before OLT	Tissue of native liver; Blood	NA	-	NA	NA	-
126	Control	Male	57	HBV Cirrhosis Recipient Before OLT	Tissue of native liver; Blood	NA	-	NA	NA	-
125	Control	Male	36	HCC Recipient Before OLT	Tissue of native liver; Blood	NA	-	NA	NA	-
124	Control	Male	57	Infection After OLT	Tissue of skin; Blood	NA	-	-	NA	-
123	Control	Male	54	HBV Cirrhosis Recipient Before OLT	Tissue of native liver; Blood	NA	-	NA	NA	-
122	Control	Male	22	PTLD After OLT	Tissue of donor liver/ileum; Blood	NA	-	-	NA	-

NA: the method was not applied on detection of this patient sample

+: HTLV-1 infection was detected

-: HTLV-1 infection was not detected

Abbreviation

HCC: hepatocellular carcinoma

OLT: orthotopic liver transplantation

AR: acute rejection

IFI: invasive fungal infection

AIH: autoimmune hepatitis

13. Table S2: Just confirming that the age range of the cohort of 400 patients ranged from a minimum of 3 months (0.25 years) to 82 years. Clinically, is it appropriate to include children and infants? For the Gender, Concomitant diseases, Preoperative diagnosis, and Collecting time, I assume these are counts? Please indicate units of measure (“n”).

We are grateful for the reviewer’s suggestion and inquiry. We have added the units of measure in Table S2. It is appropriate to include children and infants in the HTLV-1 prevalence screening. In our hospital, the recipients commonly include infants and children. HTLV-1 can be transmitted from infected mother to child during prolonged breastfeeding so that it could infect the infant recipients before surgery (Percher et al., Viruses, 2016; Prendergast et al., The Lancet Child & Adolescent Health, 2019). In addition, children and infants can also be infected with HTLV-1 during pediatric liver transplantation through blood transfusion or the donor source, like adults. Therefore, the HTLV-1 infection in infants or children is counted in the overall population infection, which has practical research value.

Table S2. Clinical information of the 400 patients in HTLV-1 screening cohort.

Characteristics of the sample	
Demographics	
male/female (n)	273/147
Age (years)	38.3 (0.25-82)
Body mass index (kg/m ²)	21 (11-31)
Concomitant diseases	
Hypertension (n)	37
Diabetes (n)	73
Preoperative diagnosis (n)	

Benign end-stage liver disease (n)	78
Malignant liver disease (n)	175
Congenital biliary disease (n)	78
Hereditary metabolic disorder (n)	11
Complications after liver transplantation (n)	31
Other diseases (n)	16
Liver donors (n)	11
	
Collecting time (n)	
	
Pre-operation (n)	149
Post-operation (n)	251

Reviewer 3

We genuinely appreciate the reviewers' gracious comments. These expert comments indeed largely facilitate our revision and increase the readability of our manuscript. We would like to provide the following point-to-point response.

Li et al., used TargetSeq to determine the presence of viral nucleic acid in various tissues derived from transplant donors associated with acute Graft vs Host Disease (aGVHD) and conclude that there is an association between HTLV and aGVHD. They then use CyTof to show that aGVHD patients have a unique immune cell subset profile in blood and that CD68+ APCs are preferentially positive for HTLV (Tax). Importantly they go on to use imaging mass cytometry to show that skin lesions also contain CD68+ APCs (likely to have migrated in from blood) and that these cells colocalise with CD4 T cells which are also Tax+. Finally, they show CD68+ Tax+ cells in liver. Overall, this study represents a significant paradigm shift in our understanding of aGVHD and this study is of high interest. However, many of the results are quite suggestive and not definitive in its current form.

1. I also found this manuscript very confusing to read indeed and I was not been able to interpret the data as the authors have. There are numerous grammatical errors throughout the manuscript that should be corrected. This made it difficult to understand the manuscript in places. There is also some major misalignment of Figures and tables. For example, the authors direct the reader to Table S3 for the CyTof panle but this is in fact listed in Table S4.

We are grateful for the reviewers' suggestion and apologize for the unclear statement in the original manuscript. Based on the reviewer's comments, the manuscript has been carefully re-written and figures have been re-organized. We have proofread our manuscripts carefully and had our manuscripts reviewed by a native speaker to avoid misleading statements.

2. Figure 2: I am especially confused by the message of Figure 2. At the beginning of the results section the authors do not tell us what Group A, B or C are and this is also

not defined in the Figure legends of Figures 1 or 2. HTLV was only detected in the 6 group A donors but not in any of the 31 donors in group B or C. Basically the authors are saying that a wide assortment of viruses were detected in a wide range of tissues (inconsistent between donors) from a wide range of donors. They then say “particularly, HTLV-1 infections were detected in none of samples from the control groups but in all the samples from aGVHD patients”. Where is this data – am I missing something here?

We are grateful for the reviewer’s inquiry. We apologize for the unclear statement in the original manuscript. To facilitate a better understanding, we have re-written the descriptions and legends of Figures 1 and 2 (**line 230-246, line 250-254, line 258-264, and line 278-280**). Our study recruited 24 recipients to study aGVHD and 400 patients to investigate the prevalence of HTLV-1 infection in our center. In the original manuscript, 24 DCD (donor after circulatory death) recipients recruited in our study were divided into group A, B, C, and D. Group A were comprised of 3 aGVHD patients with almost simultaneous onset in September 2017, while Group D included 4 aGVHD patients with onset in other years. Group B was the post-transplant control group (N=11) and was further divided into 3 subgroups: a post-transplant rejection subgroup (B1, N=4), a post-transplant infection subgroup (B2, N=3), and a post-transplant normal recovery subgroup (B3, N=4). Group C was the pre-operative control group (N=6). For ease of understanding, we have simplified the grouping principles in the revised manuscript. The 24 DCD recipients are now divided into two groups, namely the aGVHD group (n=7) and the control group (n=17) (Figure 1). The blood and tissue samples from 7 aGVHD patients were tested by TargetSeq, CyTOF, ELISA, and IMC, as listed in Table S1. The detailed grouping principles and acquired samples are described in the method and materials, the legend of figures and the results (**line 116-123, line 128-142, line 230-246, line 250-254, line 258-264, and line 278-280**).

Figure 1. A schematic of the experimental design. (A) Multiple viral DNA and RNA sequences were detected in 37 specimens from the patients in the aGVHD group (N=2) and the control group (N=17). Tested specimens included skins with rash, peripheral blood, liver grafts, intestines, ventricular walls, and lungs. (B) CyTOF was performed on a total of 16 samples of peripheral blood mononuclear cells (PBMCs) from the patients in the aGVHD group (N=3) and the control group (N=10). (C) 17 FFPEs from the patients in the aGVHD group (N=7), including 7 skin tissues, 5 native liver tissues, 3 graft liver tissues, and 2 bone marrow tissues, were assessed with IMC.

The detection results of HTLV-1 infection were showed in Figure 2A. Figure 2 depicted the subsequent result of TargetSeq detection. 6 samples, including peripheral blood, skins, or liver biopsy tissues, from only 2 aGVHD recipients (ID 139 and 141) were acquired (Figure 2A). Each square represented a detection result for each virus in each sample. Each row represented the detection result of each virus and each column showed 37 samples from 19 recipients in the GVHD and control groups. The top of the heatmap showed the patients' ID and group, and the bottom colors indicate the tissue source of detected samples. Yellow, red, and blue squares respectively indicate the sample is infected, integrated, and uninfected by the virus. Therefore, HTLV-1 was detected in all samples from the aGVHD group, but not in 31 samples from the other groups.

Figure 2. Multiple viral cDNA probe detections indicate that HTLV-1 presents exclusively in all the peripheral blood and tissues from aGVHD patients. (A) The heatmap depicted detection of virus or virus/host integration for each sample from two patients in the aGVHD group and 17 patients in control group. Negative, positive, and host integration results are respectively marked as blue, yellow, and red. (B) Patient numbers of virus positive detection and host integration were counted. Every virus with positive detection exhibited host gene integration. (C) The percentages of virus infection rates in each group were calculated and compared between the aGVHD group and control group. HTLV-1 was detected in all six collecting samples from two patients in aGVHD group, but in none of the samples from patients in the control group. all p-values were calculated using Fisher's exact test and corrected with Benjamini-Hochberg adjustment. * $p < 0.05$. (D) The percentage of virus-infected patients with no detectable virus in tissues or peripheral blood were calculated. (E) The heatmap depicts basic clinical information of the 400 patients in the validation cohort, which included age, body mass index (BMI), sample collection time, preoperative diagnosis, and concomitant disease. Benign ESLD: Preoperative diagnosis includes benign end-stage liver disease, Malignant LD: malignant liver disease, Complications after LT: Complications after liver transplantation.

complications after liver transplantation, NC: no concomitant disease. Red arrows with annotations of the viral loads marked two HTLV-1 infection patients.

Line 116-123: The DCD recipients were divided into two groups, the aGVHD and control groups. The aGVHD group comprised 7 patients diagnosed with aGVHD in different years (2015, 2017, 2018, and 2020). The control group consisted of 17 recipients, including post-transplant rejection patients (N=4), post-transplant infection patients (N=3), post-transplant regular recovery recipients (N=4), and pre-operative patients (N=6). For patients with aGVHD, post-transplant rejection, or post-transplant infection, samples were collected during the disease progression. Specimens were collected before surgery for preoperative control patients.

Line 128-142: For TargetSeq assay, fresh samples, including skin affected by a rash, peripheral blood, and biopsy tissues of livers, intestines, ventricular walls, and lungs, were harvested from 2 patients in the aGVHD group (ID 139 and ID141) and 17 recipients in the control group (Table S1).

For CyTOF and subsequent analysis, the PBMCs samples were collected from 3 aGVHD patients in the aGVHD group (ID 139, ID 140, and ID 141) and 11 recipients of the control group, including post-transplant rejection patients (N=4), post-transplant infection patients (N=3), and post-transplant regular recovery recipients (N=4). The PBMCs of the aGVHD group were collected at two different time points between onset and death. The PBMCs in the control group were collected only once.

The specimens for IMC were formalin fixed and paraffin embedded (FFPE) sections of native livers, skin lesions, liver biopsies (graft tissues), and bone marrows obtained from the 7 aGVHD patients (Table S1). The FFPE sections of native livers were collected during transplant surgeries, and the other specimens were collected during the progression of aGVHD.

Line 230-246: To reveal the cryptogenic infections associated with the development of aGVHD, TargetSeq was adopted to detect viral infection in samples from 2 patients (ID

139 and ID141) in the aGVHD group and 17 recipients in the control group (Fig. 1A). Detailed patient information and grouping principles were described in the Method and materials. In the aGVHD group, the peripheral blood, skins, or liver biopsy tissues from only 2 patients were acquired due to limits of sample collection (Fig. 2A). Likewise, peripheral blood and disease-related lesion tissue specimens, including liver tissues, infectious site tissues, and lung metastases of liver tumors, were collected from the control group (Fig. 2A and Table S1). TargetSeq allowed us to determine the existence of viruses (HHV-6A, HTLV-1, HTLV-2, HTLV-4, EBV, adenovirus, CMV, and parvovirus-1), especially whether viruses were in the latent infection phase (Fraser et al., Science, 2014). Latent infection, where viral DNA had been integrated into the host genome, is a way for viruses to achieve persistent and lifelong infection in humans and participate in the pathogenesis of various complications (Bangham, Annu Rev Immunol, 2018). In the aGVHD and control groups, HTLV-2 and HTLV-4 were not detected and therefore excluded from subsequent analysis (Fig. 2A). HHV-6A and HTLV-1 were detected in all the aGVHD samples tested, while EBV and adenovirus were detected in some of the aGVHD samples (Fig. 2A).

Line 250-254: The presence of viruses was compared between groups by Fisher's exact test and corrected with Benjamini-Hochberg adjustment (Fig. 2C). Compared with the control group, the percentage of HTLV-1 positive patients was significantly higher in the aGVHD group (adjusted p-value = 0.0406) (Fig. 2C).

Line 258-264: Although HHV-6A was detected in all samples from the aGVHD group, HHV-6A was also present in the control group. There was no significant difference in the HHV-6A infection rates between the aGVHD and control groups (Fig. 2A and 2C). Therefore, these results preliminarily indicated the correlation between HTLV-1 infection and the occurrence of aGVHD. In addition, some of the viruses were not detected in the peripheral blood of the patients with tissue infections, which suggest it may be more reliable to perform virus screening on both tissue and peripheral blood (Fig. 2D).

Line 278-280: Given that, the possibility that the co-occurrences of the two rare diseases were a coincidence was less than 10^{-4} using a binomial probability calculation, suggesting that HTLV-1 infection should be closely related to aGVHD development after LT.

Table S1 Patients' information profile.

ID	Group	Sex	Age	Diagnosis	Sample type	HTLV-1 detection method				HTLV-1 detection summary
						IMC	Targetseq	CyTOF	ELISA	
145	GVHD	Male	41	aGVHD After OLT	FFPE of skin, native liver; Blood	+	NA	NA	+	+
144	GVHD	Female	62	aGVHD After OLT	FFPE of skin, native liver; Blood	+	NA	NA	+	+
143	GVHD	Female	47	aGVHD After OLT	FFPE of skin	+	NA	NA	NA	+
142	GVHD	Male	50	aGVHD After OLT	FFPE of skin	+	NA	NA	NA	+
141	GVHD	Female	54	aGVHD After OLT	Tissue/FFPE of skin, native/donor liver; Blood	+	+	+	NA	+
140	GVHD	Male	45	aGVHD After OLT	Tissue/FFPE of skin, native/donor liver; Blood	+	NA	+	NA	+
139	GVHD	Female	63	aGVHD After OLT	Tissue/FFPE of skin, native/donor liver; Blood	+	+	+	NA	+
138	Control	Male	48	Normal recovery After OLT	Tissue of donor liver; Blood	NA	-	-	NA	-
137	Control	Male	59	ACR After OLT	Tissue of donor liver; Blood	NA	-	-	NA	-
136	Control	Male	23	HCC recurrence After OLT	Tissue of lung; Blood	NA	-	-	NA	-
135	Control	Female	31	IFI After OLT	Tissue of heart; Blood	NA	-	-	NA	-
134	Control	Male	60	AIH Recipient Before OLT	Tissue of native liver; Blood	NA	-	NA	NA	-
133	Control	Female	32	ACR After OLT	Tissue of donor liver; Blood	NA	-	-	NA	-
132	Control	Male	42	ACR After OLT	Tissue of donor liver; Blood	NA	-	-	NA	-
131	Control	Male	37	Normal recovery After OLT	Tissue of donor liver; Blood	NA	-	-	NA	-

130	Control	Male	52	HCC Recipient Before OLT	Tissue of native liver; Blood	NA	-	NA	NA	-
129	Control	Female	46	ACR After OLT	Tissue of donor liver; Blood	NA	-	-	NA	-
128	Control	Male	56	New hepatitis B After OLT	Tissue of donor liver; Blood	NA	-	-	NA	-
127	Control	Male	67	HCC Recipient Before OLT	Tissue of native liver; Blood	NA	-	NA	NA	-
126	Control	Male	57	HBV Cirrhosis Recipient Before OLT	Tissue of native liver; Blood	NA	-	NA	NA	-
125	Control	Male	36	HCC Recipient Before OLT	Tissue of native liver; Blood	NA	-	NA	NA	-
124	Control	Male	57	Infection After OLT	Tissue of skin; Blood	NA	-	-	NA	-
123	Control	Male	54	HBV Cirrhosis Recipient Before OLT	Tissue of native liver; Blood	NA	-	NA	NA	-
122	Control	Male	22	PTLD After OLT	Tissue of donor liver/ileum; Blood	NA	-	-	NA	-

NA: the method was not applied on detection of this patient sample

+: HTLV-1 infection was detected

-: HTLV-1 infection was not detected

Abbreviation

HCC: hepatocellular carcinoma

OLT: orthotopic liver transplantation

AR: acute rejection

IFI: invasive fungal infection

AIH: autoimmune hepatitis

3. They go on to say “Given the overall positive rate of HTLV-1 in all tested samples was 6/37, the possibility (p-value) that it was a coincidence was less than 10^{-7} , which was practically preclusive”. I do not understand how detecting HTLV in only 6 of 37 donors is significant? There seems to be nothing special about HTLV presented in

Figure 2. I am sure I must be missing something. This section needs to be rewritten if this reviewer is going to understand it.

We are grateful for the reviewer's suggestion and inquiry. We apologize for the unclear statement in the original manuscript. We have re-written the manuscript to make it easy to understand. Figure 2 depicted the result of TargetSeq detection, allowing us to determine the existence of viruses (HHV-6A, HTLV-1, HTLV-2, HTLV-4, EBV, adenovirus, CMV, and parvovirus-1) (Figure 2A). TargetSeq was performed on 6 samples from 2 aGVHD group patients and 31 samples from 17 recipients in control group. HTLV-1 infection was only detected in 6 samples from aGVHD group and none sample from control group. We used binomial probability calculation to calculate the possibility that HTLV-1 infection detected in all samples from aGVHD patients was coincident. The calculation result of 10^{-7} indicates the probability of this being by chance is extremely low ($(\frac{6}{37})^6 \times (\frac{31}{37})^{31} \approx 7.5 \times 10^{-8}$, where 37 = the number of all samples; 6 = the number of HTLV-1 positive samples).

Figure 2. Multiple viral cDNA probe detections indicate that HTLV-1 presents exclusively in all the peripheral blood and tissues of aGVHD patients. **(A)** The heatmap depicts the detection of virus or virus/host integration in each sample from 2 patients in the aGVHD group and 17 patients in the control group. Negative, positive, and host integration results are marked as blue, yellow, and red, respectively.

To avoid artificially yielding positive results, we have modified the manuscript to replace the sample infection rate with the patient infection rate and re-performed the statistical analysis (**line 250-254**), as Reviewer 2 suggested. According to the cohort of 400 patients, the probability of 1 patient being HTLV-1 in our center is 0.005. Two among nine-teen patients were detected to be HTLV-1 positive by TargetSeq. Therefore, the possibility (p-value) of detecting HTLV-1 in all aGVHD patients but no control patients as coincidental is less than 10^{-4} using a binomial probability calculation ($0.005^2 \times (1 - 0.005)^{19-2} \approx 2.3 \times 10^{-5}$) (**line 278-280**).

After simplifying grouping principles, the Fisher's exact test is re-performed to compare the infection patient counts of six viruses between the aGVHD and control groups. TargetSeq did not detect HTLV-2 and HTLV-4 in the aGVHD and control groups, so they are excluded from subsequent analysis (Figure 2A). When comparing the aGVHD to control group (HTLV-1 infection), a Fisher's exact test of 2/2 (100%) vs. 0/17 (0%) is performed, resulting in a p-value of 0.0058 (Figure 2C). Since Fisher's exact tests are performed 6 times, p-values are corrected p-values with Benjamini-Hochberg adjustment, using the formula $q\text{-value} = p\text{-value} \times \frac{n}{k}$, $n =$ numbers of Fisher's exact test ($k =$ the individual p-value's rank, $q\text{-value} =$ adjusted p-value). Compared with the control group, the percentage of HTLV-1 positive patients is significantly higher in the aGVHD group with a q-value of 0.0406 ($0.0056 \times \frac{6}{1}$) (**line 230-246** and **line 250-254**). Therefore, correlation between HTLV-1 infection and aGVHD occurrence was preliminarily revealed by TargetSeq.

Figure 2. Multiple viral cDNA probe detections indicate that HTLV-1 presents exclusively in all the peripheral blood and tissues of aGVHD patients. **(C)** The percentages of virus infection rates in each group were calculated and compared between the aGVHD and control groups. HTLV-1 was detected in all the 6 samples from the 2 patients in the aGVHD group, but none of the samples from the control group. All p-values were calculated using Fisher's exact test and corrected with Benjamini-Hochberg adjustment. * $p < 0.05$.

Line 230-246: To reveal the cryptogenic infections associated with the development of aGVHD, TargetSeq was adopted to detect viral infection in samples from 2 patients (ID 139 and ID141) in the aGVHD group and 17 recipients in the control group (Fig. 1A). Detailed patient information and grouping principles were described in the Method and materials. In the aGVHD group, the peripheral blood, skins, or liver biopsy tissues from only 2 patients were acquired due to limits of sample collection (Fig. 2A). Likewise, peripheral blood and disease-related lesion tissue specimens, including liver tissues, infectious site tissues, and lung metastases of liver tumors, were collected from the control group (Fig. 2A and Table S1). TargetSeq allowed us to determine the existence of viruses (HHV-6A, HTLV-1, HTLV-2, HTLV-4, EBV, adenovirus, CMV, and parvovirus-1), especially whether viruses were in the latent infection phase (Fraser et

al., Science, 2014). Latent infection, where viral DNA had been integrated into the host genome, is a way for viruses to achieve persistent and lifelong infection in humans and participate in the pathogenesis of various complications (Bangham, Annu Rev Immunol, 2018). In the aGVHD and control groups, HTLV-2 and HTLV-4 were not detected and therefore excluded from subsequent analysis (Fig. 2A). HHV-6A and HTLV-1 were detected in all the aGVHD samples tested, while EBV and adenovirus were detected in some of the aGVHD samples (Fig. 2A).

Line 250-254: The presence of viruses was compared between groups by Fisher's exact test and corrected with Benjamini-Hochberg adjustment (Fig. 2C). Compared with the control group, the percentage of HTLV-1 positive patients was significantly higher in the aGVHD group (adjusted p-value = 0.0406) (Fig. 2C).

Line 278-280: Given that, the possibility that the co-occurrences of the two rare diseases were a coincidence was less than 10^{-4} using a binomial probability calculation, suggesting that HTLV-1 infection should be closely related to aGVHD development after LT.

4. Figure 3: The authors show that PBMC derived from HTLV+ patients show a different immune cell profile to HTLV- patients and the many HTLV+ (Tax+) cells express CD68 which is a marker on monocytes in blood and macrophages in tissue. Although I agree with the concluding remark of the results describing the figure "Thus, CyTOF results revealed aGVHD specific immune cell subsets, especially Tax+ CD68+ APC cells" I do not agree with way the authors define cell population in blood – especially macrophages which are tissue resident cells! Furthermore, macrophages are very weak antigen presenting cells. The authors claim to have identified macrophages as CD68+CD11c+ HLA-DR+ cells in peripheral blood. This does not bode well for the authors understanding of the mononuclear phagocyte system as macrophages are not present in blood and do not express CD11c. These cells are more likely a NK or DC population.

We are grateful for the reviewer's comments and suggestion. We do agree with the reviewer's discussions regarding the mononuclear phagocyte system. We apologize that we had mistakenly defined the seven Tax⁺CD68⁺ clusters in the original manuscript. We assumed the disease specific CD68⁺ cells were induced by extreme pathological condition and mistakenly identified them as macrophages based on CD68 expression. After carefully studying related literatures, we have modified our definition of seven Tax⁺CD68⁺ clusters and added expression levels of CD3, CD19, CD14, CD16, CD11c, CD123, and HLA-DR for more precise classification (Table S5). Cluster 29# (CD3⁻CD19⁻CD11c⁺CD123⁻HLA-DR⁺), 25# (CD3⁻CD19⁻CD14⁺CD11c⁺CD16⁺) and 26# (CD19⁻CD14⁻CD3⁺CD8⁺Tax⁺) were respectively cDCs, intermediate monocytes, and Tax⁺CD8⁺T cells (Iberg et al., 2017; Ziegler-Heitbrock et al., 2010).

Our CyTOF panel did not include CD56 and CD66b, so we might not directly identify natural killer (NK) cells or neutrophils. Cluster 22#, 24#, 23#, and 30# were not classical monocytes, T cells, B cells or DCs. Yet, since cluster 22#, 23#, and 30# have biomarker expressions as CD3⁻CD19⁻CD14⁻CD16⁺HLA-DR⁻CD68⁺, these cell clusters are classified as natural killer cells or monocytes. Because clusters 22#, 23#, and 30# expressed CD68, a traditional marker of monocytes and macrophages, and were derived from peripheral blood, they are defined as non-classical monocytes. Cluster 24# does not express CD3, CD14, CD19, CD16, CD11c, or HLA-DR. We therefore do not find enough evidence to define them as NK cells or neutrophils when CD56 and CD66b were absent. Additionally, cluster 24# expresses activation markers such as CD38 (Malavasi et al., 2008) and CD69 (Ziegler et al., 1994), indicating cluster 24# was highly activated (**line 300-321**).

Original

Revised

Figure 3. Circulating Tax⁺ CD68⁺ immune cells are uncovered distinctively in aGVHD peripheral blood, mainly DCs and monocytes. (A) The two histograms visualize the intensities of HTLV-1-specific Tax signal in CD45⁺ populations (Figure S1) from the two groups of PBMC samples. In aGVHD group, PBMCs from three aGVHD patients were collected at two-time points, T1 and T2, between onset and death. Ten patients in the control group received liver transplants and their PBMCs were collected at one postoperative time point when the complication occurred. PBMCs from aGVHD patients show high levels of HTLV-1 Tax expression. (B) t-SNE dimensional reduction was performed on CD45⁺ cells to explore the heterogeneity across patients. The t-SNE maps show their expression profiles. The color scale indicates the arcsinh-transformed signal intensity of Tax. The clusters with positive Tax expressions were marked through manual gating (red circles), primarily presented in the aGVHD group. (C) PhenoGraph and t-SNE depict a clustering result with 30 clusters identified automatically. The aGVHD group displays distinct cluster distributions from the control group. The Tax⁺ subpopulations were marked with hollow red circles (Table S5). (D) A heatmap depicts the median expression of cell markers in the seven Tax⁺ subpopulations. The HLA-DR⁺ subpopulations were marked with a hollow red rectangle. The color bars indicate arcsinh-transformed signal intensities of the proteins. (E) Percentages of the seven Tax⁺

subpopulations in CD45⁺ cells were compared between the aGVHD and control groups. Tax⁺ cells are presented mainly in the aGVHD group. (Intermediate monocyte: iMo; Non-classical monocyte: ncMo; Conventional dendritic cell: cDC; CD8⁺ T cell.)

Table S5. Characteristic phenotype of aGVHD group specific clusters.

Cluster	No.22	No.23	No.24	No.25	No.26	No.29	No.30	
Phenotype (+)				Tax				
				CD45				
				CD183				
				CD95				
				CD9				
				CD68				
		CD16	CD11c	CD194	CD11c	CD11c	CD11c	CD11c
		CD194	CD127	CD197	CD14	CD194	CD14	CD16
		CD21	CD16	CD38	CD127	CD3	CD194	CD194
		CD39	CD194	CD58	CD16	CD8	CD30	CD197
		CD38	CD38	CD69	CD194	CD38	CD38	CD21
			CD39	IgM	CD197	CD39	CD39	CD30
			CD58		CD21	HLA _DR	HLA _DR	CD39
			CD69		CD39	CD5	CD58	CD58
				CD58	IgM	CD69	IgM	
				IgM		CD8a		
Summary	nMo	ncMo	unknown	iMo	CD8 ⁺ T cell	cDC	nMo	

Abbreviation

iMo: intermediate monocyte

cDC: dendritic cell

nMo: non-classical monocyte

Line 300-321: In Figure 3B, every dot represents a single cell, and color denotes the expression level of Tax protein. The spatial distributions of the immune cells were different between the aGVHD and control groups, suggesting different immune responses in these patients (Fig. 3B). We then clustered immune cells into 30 clusters based on the surface marker expressions with PhenoGraph(Chen et al., PLoS Comput Biol, 2016) (Fig. 3C). The major different clusters between the aGVHD and control groups were Tax⁺CD68⁺ clusters (Fig. 3B and S2). In consistent with the result of TargetSeq that HTLV-1 infection should be closely related to aGHVD development

after LT, Tax⁺ cells were only detected in cells from the aGVHD group, which composed the clusters 22-26# and 29-30# (Fig. 3B and 3C). The cell phenotypes of the Tax⁺ clusters are listed in Table S5. Cluster 23#, 25#, 29# and 30# were classified as CD3⁻CD19⁻CD11c⁺CD123⁻ cells. Among them, Cluster 29# were HLA-DR⁺ cells, which can be defined as conventional dendritic cells (cDC cells). Dendritic cells (DCs), constituting the mononuclear phagocyte system (MPS), are considered to be the most potent APCs (Iberg et al., Trends in Immunology, 2017). Because CD68 is a selective marker for human monocytes and macrophages, Clusters 22#, 23# and 30# (CD3⁻CD19⁻CD14⁻CD16⁺HLA-DR⁻CD68⁺) were considered non-classical monocytes (Iqbal et al., Blood, 2014). Cluster 25# was intermediate monocytes (CD3⁻CD19⁻CD14⁺CD11c⁺CD16⁺CD68⁺). Cluster 24# (CD3⁻CD19⁻CD14⁻CD16⁻CD11c⁻CD123⁻HLA-DR⁻CD68⁺) expressed activation markers such as CD38 (Malavasi et al., Physiol Rev, 2008) and CD69 (Ziegler et al., Stem Cells, 1994), indicating these cells were highly activated. Meanwhile, cluster 26# (CD19⁻CD14⁻CD3⁺CD8⁺Tax⁺) was characterized as Tax⁺CD8⁺T cells (Fig. 3C, 3D and S2). Phenograph clustered 10 to 100 times more cells from the aGVHD group than the control group into the seven Tax⁺ clusters (Fig. 3E).

5. Figure 4-6. The use of imaging mass cytometry is a clear strength of this manuscript. However, it is a shame that some key APC markers are missing. It would have been useful to include some more macrophage specific markers such as FXIIIa and also DC markers such as CD1c. There are no markers to include cDC1 (e.g. XCR1, CADM1, CLEC9A). Recently, CD14⁺ tissue cells have been shown to consist of monocyte derived macrophages (CD14⁺ CD1c⁻ CD11c⁻) and monocyte derived dendritic cells (CD14⁺ CD1c⁺ CD11c⁺). Both of these express CD68 – PMID 33846309. As DCs are more potent APCs than macrophages it would be helpful for the authors to define if the APCs they define in skin express CD11c (which is included in the IMC panel) and discern if they are MDM or MDCC.

We are grateful for the reviewer's comments and suggestions. To address the reviewer's comments, we have corrected the comprehensive phenotypes of APCs based on

literature reports and revised the manuscript and Figures 4-6 accordingly (**line 380-403**). The original Figure 5 is moved into supplementary information (Figure S6) and identification of CD68⁺ mononuclear phagocytes (MNPs) in skin sections of aGVHD patients is presented in the revised version of Figure 5. In Figure 5, CD11c, CD14, CD123, CD68, HLA-DR are applied to defined APC phenotypes. Therefore, CD11c⁺HLA-DR⁺CD68⁺, CD123⁺HLA-DR⁺CD68⁺, CD14⁺CD11c⁻CD68⁺, and CD14⁻CD11c⁻CD68⁺ cells are defined respectively as can be clearly defined as conventional DCs (cDCs) and plasmacytoid DCs (pDCs), monocyte-derived macrophages (MDMs), and macrophages, respectively (Kashem et al., Annual Review of Immunology, Vol 35, 2017; Mair and Liechti, Cytometry Part A, 2021; Rhodes et al., Nat Commun, 2021). Moreover, we still found Tax protein expression on APCs, which helped us partly understand the rapid spread of HTLV-1 and the transduction of immune signaling (Figure 4D and 4H).

We regret that the absence of some APCs markers, such as FXIIIa, XCR1, CADM1, and CD1c, hindered further subdivision of APCs phenotypes. Our study was primarily designed to explore the causative factor of aGVHD after liver transplantation, so we mainly focused on donor-derived T cells, where HTLV-1 infected donor-derived T cells were indeed found in skin sections of female aGVHD patients. However, it is found in our results that APCs play a significant role in peripheral blood and skin lesions. Therefore, it also pointed out the direction for our future research.

Line 380-403: After identification of Tax⁺ CD68⁺ cells in peripheral blood and skin sections, the characteristics of APCs in skin lesions of all aGVHD patients were investigated by IMC to further reveal the role of HTLV-1 infection in aGVHD. The primary APCs in the skin are tissue mononuclear phagocyte (MNP) subsets, which are two prominent families, DCs and tissue-resident macrophages(Kashem et al., Annual Review of Immunology, Vol 35, 2017). DCs are further divided into conventional DCs (cDCs) and plasmacytoid DCs (pDCs). cDCs, pDCs, monocyte-derived macrophages (MDMs), and tissue-resident macrophages respectively express CD11c⁺HLA-DR⁺CD68⁺, CD123⁺HLA-DR⁺CD68⁺, CD14⁺CD11c⁻CD68⁺, and CD14⁻CD11c⁻

CD68⁺ cells (Mair and Liechti, *Cytometry Part A*, 2021; Rhodes et al., *Nat Commun*, 2021). The phenotypes and distributions of the four MNPs found by IMC in the skin lesions were presented by three representative patients (ID139, ID140, and ID145, Figure 5A-C). Single color images of HLA-DR, CD14, CD123, CD68, and CD11c from the seven GVHD patients were shown with nuclear signals in Figure S7 and S8. The signals of DNA, HLA-DR, CD14, CD123, CD68, CD11c, and the merged panels of these signals are shown in Figure 5A-C. The summarized MNPs phenotypes of all the seven aGVHD patients are shown in Figure 5D. Macrophages existed in skin sections of all the seven aGVHD patients, and MDMs were found in four patients (ID139, ID141, ID 142, ID144), while cDCs and pDCs only presented in skin sections of two patients (ID140 and ID143). In addition, some of the MNPs expressed the Tax protein (Figure 4D and 4H), suggesting that they take up HTLV-1 and transmit the virus and the signal outward. Thus, MNPs cells found in aGVHD patients were diverse, mainly DCs and macrophages, and some of them were Tax⁺CD68⁺ cells, consistent with the aGVHD-specific cell subsets in peripheral blood. The IMC results of skin sections reproduced the antigen-presenting responses and presented the APCs which were previously detected in peripheral blood and possibly migrated into the tissues.

Original

Revised

Figure 5. CD68⁺ MNPs in skin sections of aGVHD patients. Representative IMC images of skin tissues from 3 aGVHD patients (ID 139, 140, and 145) showed the 7 single signals of (A) DNA (blue), HLA-DR (red), CD11c (purple), CD14 (yellow), (B) CD123 (light blue), CD68 (green). (C) overlay signal of 7 markers. The same markers were shown in images in each column. Red, yellow, green, and blue arrows respectively pinpointed the monocyte-derived macrophages (MDMs), macrophages, plasmacytoid dendritic cells (pDCs), and conventional dendritic cells (cDCs). Red, yellow, green, and blue boxes respectively showed the magnified multi-signal overlap position of MDMs, macrophages, pDCs, and cDCs. Scale bar =100 μ m for landscape view and scale bar =20 μ m for magnified view. (D) Each column is mononuclear phagocyte (MNP) identification in skin sections from each patient. Each row is the comprehensive phenotyping of one MNP, and the symbol +/- indicates the MNP is existence/absent in skin sections of each patient. Displayed channels for each row are: macrophage (CD14⁺CD11c⁻CD68⁺); MDM (CD14⁺CD11c⁻CD68⁺); cDCs (CD11c⁺HLA-DR⁺CD68⁺); pDCs (CD123⁺HLA-DR⁺CD68⁺).

6. It is of note that DCs and macrophages both express CD4, so this combination of marker expression cannot be used to definitively define T cells. The authors need to show these cells are negative for APC markers (e.g. CD11c, CD14, CD68 and HLA-DR also included in the panel).

Thanks a lot for the suggestion. We apologized for unclear definition of T cells in IMC and subsequent analysis. According to the reviewer's suggestion, we have added magnified views of "Triple positive T cells" in Figure 4 and overlay signals of CD3 and CD4 signal are presented in Figure S5 to identify CD4⁺ T cells. We have also added single color images of CD11c, CD14, CD68 and HLA-DR with nuclear signals in Figure S7 and S8 to show these cells are negative for APC markers.

Figure 4. HTLV-1 specific Tax protein detected on the aGVHD skin tissues by IMC. Representative IMC images of the skin tissues from 7 aGVHD patients showed the overlap of Tax (red), DNA (blue) and different immune response markers. Displayed channels for each column were: (A) Tax (red)/DNA (blue); (B) CD4 (green)/Tax (red)/EIF1AY (white)/DNA (blue); (C) C4d (green)/ Tax (red)/ EIF1AY (white)/DNA (blue); (D) CD68 (green)/Tax (red)/ EIF1AY (white)/DNA (blue);

(D) CD68 (green)/ Tax (red)/EIF1AY (white)/ DNA (blue). (E) Tax (red)/DNA (blue); (F) CD4 (green)/ Tax (red)/ DNA (blue); (G) C4d (green)/ Tax (red)/ DNA (blue); (H) CD68 (green)/ Tax (red)/ DNA (blue). Red arrows pinpointed the multi-signals overlap position and red box showed the magnified view of CD4⁺T cells with Tax signal. Scale bar =100 μ m for landscape view and scale bar =20 μ m for magnified view.

Figure S5. CD4⁺ T cells detected on aGVHD skin tissues by IMC. Representative IMC images of skin tissues from seven aGVHD patients showed the overlap of DNA (blue) and CD3, CD4, and EIF1AY. Displayed channels for each column were: CD3 (green)/DNA (blue); CD4 (green)/DNA (blue); CD3 (green)/CD4 (red)/DNA (blue); EIF1AY (green)/DNA (blue). Red arrows pinpointed the CD3⁺CD4⁺T cells position and red box showed the magnified view. Scale bar =100 μ m for landscape view and scale bar =20 μ m for magnified view.

Figure S7. Single marker signals on aGVHD skin tissues by IMC. Representative IMC

images of skin tissues from seven aGVHD patients showed the overlap of DNA (blue) and five different markers (green), including CD68, C4d, CD11c, HLA_DR, and CD123. Scale bar =100 μ m.

Figure S8. Single marker signals on aGVHD skin tissues by IMC. Representative IMC images of skin tissues from seven aGVHD patients showed the overlap of DNA (blue) and four different markers (green), including CD8, CD19, CD16, and CD14. Scale bar =100 μm .

REVIEWERS' COMMENTS

Reviewer #1 (Remarks to the Author):

The authors had made effort to address this reviewer's concerns and no further questions.

Reviewer #2 (Remarks to the Author):

The authors have thoroughly addressed all reviewers' comments and I appreciate the effort and time this took. There are still residual problems and I think that too much additional text has been added in the results sections and captions of the figures. A content expert should weigh in on the relevance of the additional comment—I am primarily a statistical and study design reviewer.

I think the modifications the authors made to the statistical analyses make sense and address the concerns previously allayed. I especially appreciate the modifications to table S1. This is a very important table for the paper.

Major comments:

There may be redundancy between the body of the manuscript and the main paper's figure captions. Eliminate redundancies for brevity.

In the statistical analysis section, state what false discovery rate was used. Also, state that the Mann-Whitney test was used (Figure S3).

Lines 298-299: The last sentence of this paragraph—I do not understand what this means or how Figure 3A shows what is stated here. Also, in Figure 3a, I can't read the axes and the y-axis is not labeled.

Minor comments:

Line 38: replace "correlated" with "associated"

Line 29-30: edit: "... in our center are, respectively, estimated to be 0.5%..."

Line 84: fix spelling to "transplantations"

Line 85: reword to "...induce aGVHD even if HLA is mismatched..."

Line 86: Reword to "Additionally, HLA miss-matching does not correlate with ..."

Line 87: Sentence, "In HSCT, ..."—I don't know what this sentence is trying to say. Reword.

Line 92: Reword to "...masked by the severe fungal and bacterial..."

Line 94: Reword to "...recipients were identified among..."

Line 98: remove "the" from in front of "...7 aGVHD patients."

Line 100: Were the 400 patients selected a random sample or did you take all cases in a given period?

Line 108: Reword to "CD4+ T cells were increased by..."

Line 119: reword to "...aGVHD group consisted of 7 patients..."

Line 125: Reword to "...from preoperative control..."

Line 126: Reword "...evaluated in this study are listed in..."

Line 133: Reword to "...and subsequent analysis, PBMC samples were..."

Line 134: reword to "... aGVHD patients (ID 139,...)" (remove "in the aGVHD group")

Line 135: Reword to "...in the control group..."

Line 243: Reword to "...In both the aGVHD groups..."

Line 254: reword to "...were not detected in any of the aGVHD..."

Line 261: use the word "association" instead of "correlation"

Line 272: reword to "...donor being lower than the lowest..."

Line 274: remove "the" from in front of "surgery for acute liver failure."

Line 276: reword to "...in our center in patients without aGVHD was 0.5%..."

Line 277: reword to "0.1% to 1.0%."

Line 283: reword to: "Since the relationship between HTLV-1 and aGVHD is statistically supported, ..."

Line 305: I don't understand this sentence. Should the first word be "consistent" or "inconsistent"? Either way, the sentence needs to be reworded for understanding.

Line 318: reword to "indicating these cells were highly..."
Line 320: change to "PhenoGraph"
Line 322: remove comma after "leukocytes"
Line 341: reword to "of EIF1AY were shown..."
Line 342: removed the "s" at the end of "serum"
Line 371: reword to "information of HTLV-1 are..."
Line 377: report p-value as "...p-value < 0.0001"
Line 393 remove "the" after "phenotypes of all..."
Line 421: Change "correlation" to "association"
Lines 433-434: the phrase "...which will refresh the recognition of the community on aGVHD..." is awkward and should be reworded.
Table S1: footnote for "NA: the method..." should be reworded to "'NA: method not applied"
Table S2: Capitalize "Male"; add percentages to all of the counts in form 37 (xx%), clarify what "Collecting time" is, and what the units are (should not be "n").
All new text in captions for figures: make sure when you are describing what the figure is displaying that you are speaking in present tense. Currently they are mostly past tense. ("showed" should be "shows", "were shown" should be "are shown", "were" should be "are", "displayed" should be "displays")

Reviewer #3 (Remarks to the Author):

I am satisfied with the authors response to my assessment and the revised manuscript. Much of my confusion has been resolved.

I especially note the rarity 7aGVHD patients identified out of 4,023 liver transplant surgeries during 2015-2022 and congratulate the authors on their outstanding access and detection of these donors.

I have only 1 last comment/suggestion:

In Figure 5, were the authors able to identify any CD14+ CD11+ dual positive cells? These would be monocyte-derived dendritic cells (as per Rhodes et al., Nat Commun, 2021 and the Segura lab) and their inclusion would add impact to the manuscript as these are more potent antigen presenting cells than CD14+ CD11- monocyte derived macrophages.

I would like to congratulate the authors on this outstanding study.

Authors' Response to the Reviewer Comments

Manuscript ID: NCOMMS-21-40067B

Title of Paper: HTLV-1 infection of donor-derived T cells underlies aGVHD after liver transplantation

We have substantially revised our manuscript based on provided comments and suggestions. The reviewers' comments are colored in **blue**; our responses are colored in **black**, and changes made to the manuscript are colored in **red**. Together with the revised manuscript, we hope this letter adequately addresses the reviewers' inquiries. We thank the reviewers for aiding us in improving the quality of our manuscript.

Response to Comments from Reviewer 2

Reviewer 2

We genuinely appreciate the reviewers' gracious comments. These expert comments indeed largely facilitate our revision and increase the readability of our manuscript. We would like to provide the following point-to-point response.

The authors have thoroughly addressed all reviewers' comments and I appreciate the effort and time this took. There are still residual problems and I think that too much additional text has been added in the results sections and captions of the figures. A content expert should weigh in on the relevance of the additional comment—I am primarily a statistical and study design reviewer. I think the modifications the authors made to the statistical analyses make sense and address the concerns previously allayed. I especially appreciate the modifications to table S1. This is a very important table for the paper.

Major comments:

1. There may be redundancy between the body of the manuscript and the main paper's figure captions. Eliminate redundancies for brevity.

We are grateful for the reviewer's comments and suggestions. We have eliminated redundancies of the manuscript and the figure legend accordingly.

2. In the statistical analysis section, state what false discovery rate was used. Also, state that the Mann-Whitney test was used (Figure S3).

Thanks a lot for the comments. We have revised the manuscript accordingly.

Line 456-460: Fisher's exact test was adopted to test the differences in HTLV-1 positive rate between groups, and Benjamini-Hochberg adjustment was applied to control the false discovery rate. The adjusted p-value <0.05 was considered statistically significant. In Fig. S3, the differences in HTLV-1 positive rates between groups were calculated by the Mann-Whitney test.

3. Lines 298-299: The last sentence of this paragraph—I do not understand what this means or how Figure 3A shows what is stated here. Also, in Figure 3a, I can't read the axes and the y-axis is not labeled.

Thanks a lot for the inquiry. The labels of x- and y-axis were added per the reviewer's suggestion. In Fig. 3A, Tax protein signals in cells from the control group and aGVHD group were detected by CyTOF. The color scale indicates the signal intensity of Tax. Tax protein intensities are presented by the horizontal axis and the cell counts by the vertical axis. Therefore, the right-shifted histogram in the aGVHD group indicates that the Tax protein signal was detected in the PBMCs from patients in the aGVHD group.

4. Minor comments:

Line 38: replace “correlated” with “associated”

Line 29-30: edit: “... in our center are, respectively, estimated to be 0.5%...”

Line 84: fix spelling to “transplantations”

Line 85: reword to “...induce aGVHD even if HLA is mismatched...”

Line 86: Reword to “Additionally, HLA miss-matching does not correlate with ...”

Line 87: Sentence, “In HSCT, ...”—I don’t know what this sentence is trying to say. Reword.

Line 92: Reword to “...masked by the severe fungal and bacterial...”

Line 94: Reword to “...recipients were identified among...”

Line 98: remove “the” from in front of “...7 aGVHD patients.”

Line 100: Were the 400 patients selected a random sample or did you take all cases in a given period?

Line 108: Reword to “CD4+ T cells were increased by...”

Line 119: reword to “...aGVHD group consisted of 7 patients...”

Line 125: Reword to “...from preoperative control...”

Line 126: Reword “...evaluated in this study are listed in...”

Line 133: Reword to “...and subsequent analysis, PBMC samples were...”

Line 134: reword to "... aGVHD patients (ID 139,...)" (remove "in the aGVHD group")

Line 135: Reword to "...in the control group..."

Line 243: Reword to "...In both the aGVHD groups..."

Line 254: reword to "...were not detected in any of the aGVHD..."

Line 261: use the word "association" instead of "correlation"

Line 272: reword to "...donor being lower than the lowest..."

Line 274: remove "the" from in front of "surgery for acute liver failure."

Line 276: reword to "...in our center in patients without aGVHD was 0.5%..."

Line 277: reword to "0.1% to 1.0%."

Line 283: reword to: "Since the relationship between HTLV-1 and aGVHD is statistically supported, ..."

Line 305: I don't understand this sentence. Should the first word be "consistent" or "inconsistent"? Either way, the sentence needs to be reworded for understanding.

Line 318: reword to "indicating these cells were highly..."

Line 320: change to "PhenoGraph"

Line 322: remove comma after "leukocytes"

Line 341: reword to "of EIF1AY were shown..."

Line 342: removed the "s" at the end of "serum"

Line 371: reword to "information of HTLV-1 are..."

Line 377: report p-value as "...p-value < 0.0001"

Line 393 remove "the" after "phenotypes of all..."

Line 421: Change "correlation" to "association"

Lines 433-434: the phrase "...which will refresh the recognition of the community on aGVHD..." is awkward and should be reworded.

We are grateful for the reviewer's inquiry suggestion. We take all cases from January 2021 to June 2021. Per the reviewer's comment, we have revised the manuscript accordingly.

5. Table S1: footnote for "NA: the method..." should be reworded to "'NA: method not applied". Table S2: Capitalize "Male"; add percentages to all of the

counts in form 37 (xx%), clarify what “Collecting time” is, and what the units are (should not be “n”).

Thanks a lot for the comments. We have revised supplementary tables accordingly.

Table S2. Clinical information of the 400 patients in HTLV-1 screening cohort.

Characteristics of the sample	
Demographics	
Male/female (n)	273/147
Age (years)	38.3 (0.25-82)
Body mass index (kg/m ²)	21 (11-31)
Concomitant diseases	
Hypertension (n)	37 (9.25%)
Diabetes (n)	73 (18.25%)
Preoperative diagnosis (n)	
Benign end-stage liver disease (n)	78 (19.5%)
Malignant liver disease (n)	175 (43.75%)
Congenital biliary disease (n)	78 (19.5%)
Hereditary metabolic disorder (n)	11 (2.75%)
Complications after liver transplantation (n)	31 (7.75%)
Other diseases (n)	16 (4%)
Liver donors (n)	11 (2.75%)
Sample collection (n)	
One week before operation (n)	149 (37.25%)
Onset time of the complication after operation (n)	251 (62.75%)

6. All new text in captions for figures: make sure when you are describing what the figure is displaying that you are speaking in present tense. Currently they are mostly past tense. (“showed” should be “shows”, “were shown” should be “are shown”, “were” should be “are”, “displayed” should be “displays”)

We are grateful for the reviewer’s comments and suggestions. We have revised the manuscript accordingly.

Line 673-677: The heatmap depicts basic clinical information of the 400 patients in the validation cohort, which includes age, body mass index (BMI), sample collection time, preoperative diagnosis, and concomitant diseases. The two HTLV-1 infection patients are marked by red arrows with annotations of the viral loads.

Line 681-682: Figure 3. Circulating Tax⁺ CD68⁺ immune cells were uncovered distinctively in aGVHD peripheral blood, mainly DCs and monocytes.

Line 690-695: The clusters with positive Tax expressions are marked through manual gating (red circles), primarily presented in the aGVHD group. (C) PhenoGraph and t-SNE depict the clustering result with 30 clusters identified automatically. The aGVHD group displays distinct cluster distributions from the control group. The Tax⁺ subpopulations are marked with hollow red circles (Table S5).

Line 696-698: The HLA-DR⁺ subpopulations are marked by a hollow red rectangle. The color bars indicate arcsinh-transformed signal intensities of the proteins.

Line 704-705: Representative IMC images of the skin tissues from 7 aGVHD patients show the overlap of Tax (red), DNA (blue) and different immune response markers.

Line 710-711: Red arrows pinpoint the multi-signals overlap positions and red boxes show the magnified view of CD4⁺T cells with Tax signals.

Line 715-718: Representative IMC images of skin tissues from 3 aGVHD patients (ID 139, 140, and 145) show the 7 single signals of (A) DNA (blue), HLA-DR (red), CD11c (purple), CD14 (yellow), (B) CD123 (light blue), CD68 (green).

Line 718-721: The same markers are shown in images in each column. Red, yellow, green, and blue arrows respectively pinpoint the monocyte-derived macrophages (MDMs), macrophages, plasmacytoid dendritic cells (pDCs), and conventional dendritic cells (cDCs).

Line 733-740: Representative Tax⁺ images of liver graft and native liver show two overlap patterns. Each row of figures display the same tissue type. Displayed channels for liver graft samples from left to right are: CD68 (green)/Tax (red)/EIF1AY (white)/ DNA (blue), CD68 (green)/Tax (red)/DNA (blue), CD68 (green)/Tax (red)/EIF1AY (white)/DNA (blue). Displayed channels for native liver samples are CD68 (green)/Tax (red)/ DNA (blue). Red arrows pinpoint the multi-signal overlap position. (B) Tax⁻ IMC images of bone marrow and native liver sections show an overlap of Tax (red) and DNA (blue).

Response to Comments from Reviewer 3

Reviewer 3

We truly appreciate the reviewers' gracious comments, especially the statistical reviews. These expertise comments indeed largely facilitate our revision and increase the readability of our manuscript. We would like to provide the following point-to-point response.

I am satisfied with the authors response to my assessment and the revised manuscript. Much of my confusion has been resolved. I especially note the rarity 7aGVHD patients identified out of 4,023 liver transplant surgeries during 2015-2022 and congratulate the authors on their outstanding access and detection of these donors.

1. I have only 1 last comment/suggestion: In Figure 5, were the authors able to identify any CD14⁺ CD11⁺ dual positive cells? These would be monocyte-derived dendritic cells (as per Rhodes et al., Nat Commun, 2021 and the Segura lab) and their inclusion would add impact to the manuscript as these are more potent antigen presenting cells than CD14⁺ CD11⁻ monocyte derived macrophages.

We are grateful for the reviewer's comments. According to the literature, we searched for CD14⁺CD11c⁺ dual positive cells. However, we did not manage to identify such a cell subsets with definite expression markers of monocyte-derived dendritic cells, which may be caused by the severe skin damage in the rapid process of GVHD disease. Therefore, we present CD14⁺CD11c⁻CD68⁺ monocyte derived macrophages